



# How much do bacterial growth properties and biodegradable dissolved organic matter control water quality at low flow?

Masihullah Hasanyar[1], Thomas Romary[1], Shuaitao Wang[2], and Nicolas Flipo[1]

[1]Geosciences and Geoengineering Department, MINES ParisTech, PSL University, 35 Rue Saint-Honoré, 77300 Fontainebleau, France
[2]UMR 7619 METIS - Sorbonne Université - 4 place Jussieu - 75252 Paris, France

**Correspondence:** Masihullah Hasanyar (masihullah.hasanyar@mines-paristech.fr)

**Abstract.** Development of accurate water quality modeling tools is necessary for integrated water quality management of river systems. The existing water quality models can simulate dissolved oxygen (DO) concentration quite well during high flow and phytoplankton blooms in rivers; however, there are discrepancies during the summer low-flow season that are assumed to be due to the uncertainties related to the organic matter contribution of the model boundary conditions. Therefore, we used the

C-RIVE biogeochemical model to evaluate the influence of controlling parameters on DO simulations at low flow. Three Sobol sensitivity analyses (SA) were carried out based on a coarse model pre-analysis whose target was to develop SA scenarios providing a reduction in the number of model parameters and computation cost as well as hiding inter-parameter interactions. The parameters studied are related to bacterial (e.g., bacterial growth rate), organic matter (OM; repartition and degradation of OM into constituent fractions), and physical factors (e.g., reoxygenation of the river due to navigation and wind), whose

variation ranges are selected based on a detailed literature review. Bacterial growth and mortality rates are found to be by far the two most influential parameters, followed by bacterial growth yield. More refined SA results indicate that the biodegradable fraction of dissolved organic matter (BDOM) and the bacterial growth yield are the most influential parameters under conditions of a high net bacterial growth rate (= growth rate – mortality rate), while bacterial growth yield is independently dominant in low net growth situations. Based on the results of this study, proposals are made for in situ measurement of BDOM under a

dense and well-equipped urban area water quality monitoring network that could provide high-frequency data. The results also indicate the need for bacterial community monitoring in order to detect potential bacterial community shifts after transient events such as combined sewer overflows and post-infrastructure improvement in treatment plants. Furthermore, we discuss the integration of BDOM in data assimilation software for better estimation of BDOM contribution from boundary conditions, which would result in improved water quality modeling.

**1 Introduction**

Dissolved oxgyen (DO) has been considered the most important indicator of water quality in surface water resources (Odum, 1956; Escoffier et al., 2018) because it integrates the biological functioning of a system as well as the impact of anthropogenic forcing. It is the main variable used to evaluate river metabolism (Odum, 1956; Staehr et al., 2010; Demars et al., 2015) by comparing the gross primary production (GPP) with ecosystem respiration (ER) and defining whether an ecosystem is



autotrophic or heterotrophic based on the net ecosystem production (NEP = GPP-ER) being positive or negative, respectively (Garnier et al., 2020). Maintaining a sufficient level of DO is necessary for the overall health of rivers, not only because of the life dependency of water species (Garvey et al., 2007), but also for preventing smell and taste degradation (Bailey and Ahmadi, 2014).

The situation of rivers during low flow is of particular interest since studies have demonstrated that the river water quality
during such flow periods is more vulnerable to degradation due to lower dilution rates. This is particularly the case if the river receives organic matter load from wastewater treatment plants (WWTP) and combined sewage overflows (CSO), thereby leading to heterotrophic conditions (Seidl et al., 1998a; Even et al., 2004; Vilmin et al., 2016; Garnier et al., 2020) in the river where very low DO levels and high fish mortality can be observed. Therefore, river water quality modeling has been one of the main research interests of water quality managers and researchers ever since the use of the very first water quality model
(Streeter and Phelps, 1925) to more complex ones (Even et al., 1998; Flipo, 2005; Wang et al., 2013) aiming to identify the main determinants of DO evolution and to forecast the response of aquatic systems to human-induced pressure, in particular due to releases of the wastewater plants.

In water quality modeling studies at low flow, the QUESTOR model was applied on the Thames (UK), which demonstrated discrepancies between observed and simulated DO at low flow (Hutchins et al., 2020) and uncertainties related to benthic
respiration were revealed to be the main reason for the mismatch. The Riverstrahler model was applied at low flow on the Mosel (Germany) (Garnier et al., 1999), the Scheldt (Belgium) (Thieu et al., 2009), and the Seine (France) Garnier et al. (2020) where discrepancies were noticed between the modeled and observed DO. Yang et al. (2010) used the WASP model to estimate DO in low-flow streams and noted that the uncertainty of the model lies in the difficulty to characterize accurately the OM degradation and nitrification rates. The QUAL2E-OTIS water quality model shows similar discrepancies, which led
Bailey and Ahmadi (2014) to conduct a sensitivity analysis to identify the governing parameters on DO. The ProSe model, which will be used in this study, also has mismatches at low flow (Vilmin et al., 2018; Wang et al., 2019; Garnier et al., 2020). Moreover, Cox (2003) compares the existing water quality models on lowland rivers (flowing slowly with low DO content) and emphasizes that the existing models lack one or more of the required criteria, in particular the inability to account for the expected uncertainties. Thus, by observing the existing models in the market, it can be understood that large discrepancies
exist between DO simulations and observations during low-flow periods and that most models are not able to simulate the evolution of DO accurately. On the other hand, the uncertainties related to parameterization of (i) OM degradation kinetics and (ii) repartition of OM input from tributary rivers, WWTPs, and CSOs among dissolved and particulate pools are assumed to play a role in the discrepancies during non-bloom low-flow periods (Wang, 2019; Wang et al., 2019, 2021).

Therefore, sensitivity analysis (SA) is necessary to study the role of organic matter contributed by model boundary conditions
on the evolution of DO and river metabolism during low-flow periods via parameterization of the organic matter repartition into biodegradable fractions and its degradation by bacterial decomposition. Several applications of SA methods can be found for water quality modeling as in Nossent et al. (2011) and for DO and $NO_3$ modeling as in Bailey and Ahmadi (2014). Wang et al. (2018) summarized a list of SA applications in hydrological and water quality modeling and applied SA in contrasting hydrological and trophic contexts, where bacterial parameters were identified as the most influential in a 80-h non-bloom low-





flow period. However, the sensitivity of water quality models has not been investigated for long-term (= 45 days to be consistent with the batch test for biodegradable fractions of OM (Servais et al., 1995)) low-flow periods against new parameters accounting for boundary condition uncertainties (OM repartition and degradation) as well as against bacterial and physical parameters.

In this article, the sensitivity of C-RIVE (Vilmin et al., 2012; Wang et al., 2018), the biogeochemical module of ProSe-PA (Wang et al., 2019), is investigated against the background of the aforementioned parameters based on 45- and 5-day DO

simulations. After incorporation of new parameters to account for repartition and degradation of OM, SA is conducted on a synthetic river system mimicking the Seine. Finally, the parameters influencing the evolution of DO and those governing the degradation and repartition of OM are selected. On the basis of the results, proposals are made for better integration of the influential parameters in data assimilation where the model is coupled with observation data to make an optimal estimate of the temporal evolution of the parameters as well as to produce better simulation results (Cho et al., 2020). Finally, some

suggestions are made for water quality monitoring in urban areas in order to fulfill the modeling and monitoring requirements.

## 2 Material and methods

This section describes the forward model, the new parameterization and the bibliographically reviewed variation ranges, the SA strategy, and the simulations settings of the study. Since the main goal is to use a SA method to determine the controlling parameters that influence DO evolution during a summer low-flow period, we considered the C-RIVE model (section 2.1.1)

as the forward model of the study and identified the parameters that need to be included in the study. Then, two new sets of parameters were added to the study to account for the uncertainties related to the parameterization of OM degradation kinetics and its repartition into different constituent fractions (section 2.2). This was followed by determining the variation ranges of the introduced parameters (2.2.3).

Three different SAs were carried out on the basis of a specific SA strategy detailed in section 2.4 based on which 260K -

360K simulations were run with the forward model on a case study resembling a non-bloom low flow (section 2.3). Depending on the output of these simulations that are DO time series, the Sobol SA method (section 2.5) was implemented to determine the influential parameters. The Sobol indices were calculated up to the second order so as to observe the inter-parameter interactions in addition to their direct and total effects.

### 2.1 ProSe-PA

ProSe-PA results from the coupling of the ProSe hydro-biogeochemical model with a particle filter (Wang et al., 2019, 2021). It was developed to assimilate high-frequency observation data for a better estimation of the model parameters and an improvement in ProSe simulation results. The ProSe model is developed to simulate the hydro-biogeochemical evolution of the Seine from upstream of Paris until Poses (close to the estuary) and has been applied and validated numerous times (Even et al., 1998, 2007; Flipo et al., 2004, 2007; Polus et al., 2011; Raimonet et al., 2015; Vilmin et al., 2015b, a; Wang et al., 2021).

In this model, the river is divided into longitudinal cells of specific length, where three set of equations corresponding to the three modules of ProSe-PA are solved (hydrodynamics, transport, and biogeochemical). First, the hydrodynamic equations are





used to determine the discharge, velocity, and depth at each cell and at each time step, followed by the transport equations for advection, dispersion, and diffusion, and finally the biogeochemical RIVE model equations for the concentrations of all the dissolved and particulate matter.

### 2.1.1 C-RIVE Biogeochemical model

C-RIVE is a C ANSI library that implements RIVE (Billen et al., 1994) concepts. It is the biogeochemical module of ProSe-PA (Wang et al., 2019), which simulates the cycles of carbon, $O_2$, and other nutrients both in the water and sediment columns of the river. The exchange of dissolved and particulate material between the two layers occurs through diffusion (Boudreau, 1997) and sedimentation-resuspension (due to shear flow and river navigation) (Martin, 2001; Even et al., 2004; Vilmin et al., 2015b), respectively. Numerous applications of RIVE exist for the ProSe and RIVERSTRAHLER softwares (See for instance Garnier et al. (1995, 2005); Even et al. (1998); Flipo et al. (2004, 2007); Thieu et al. (2009); Vilmin et al. (2016); Marescaux et al. (2020)).

RIVE simulates the macro-nutrients cycles (C,N,P,$O_2$) based on the physiology of micro-organisms living in water and/or sediments (heterotrophic bacteria, nitrifying bacteria, and phytoplankton), the kinetics of underlying physical-chemical processes, and carbon and nutrient inputs. Model parameters (a hundreds to a few hundreds depending on the number of mico-organisms' species simulated) are mostly determined through experiments. Wang et al. (2018) summarized the list of physical, bacterial, and phytoplanktonic parameters related to the carbon cycle with their variation ranges. We examine the equations for DO and OM evolution to understand the role of the different parameters and to select the appropriate parameters for inclusion in the SA.

### 2.1.2 Dissolved oxygen evolution equations

DO in the water column (Fig. 1) depends on physical, bacterial, and phytoplanktonic processes:

$$\frac{d[O_2]}{dt} = \frac{d[O_2]}{dt}_{physical} + \frac{d[O_2]}{dt}_{phytoplanktonic} + \frac{d[O_2]}{dt}_{bacterial} \tag{1}$$

The physical process depends on reaeration due to dams, wind, navigation, the oxygen holding capacity of water, and the diffusion of oxygen between the water-sediment interface as follows:

$$\frac{d[O_2]}{dt}_{physical} = \frac{K_{rea}}{h}([O_2]_{sat}(T) - [O_2]) - \frac{D_s}{h}([O_2]_{water} - [O_2]_{sed}) + \frac{d[O_2]}{dt}_{dams} \tag{2}$$

where,

$h$: water depth [m]

$[O_2]_{sat}(T)$: the saturated oxygen concentration at temperature T $[mgO_2/L]$

$D_s$: the coefficient of diffusion between water and sediment layer $[m/s]$

$K_{rea}$ : the reoxygenation coefficient calculated from the empirical formula of Thibodeaux et al. (1994) as follows:

$$K_{rea} = \sqrt{\frac{D_m V_{wat}}{h}} + (K_{wind}V_{wind}^{2.23}(D_m * 10^4)^{2/3} + K_{navig}) \tag{3}$$





where,

$K_{wind}$: reoxygenation coefficient due to wind $[m/s]$

$V_{wind}$: wind speed $[m/s]$

$V_{wat}$: river flow velocity $[m/s]$

$K_{navig}$: reoxygenation coefficient due to navigation $[m/s]$ (Vilmin, 2014)

$D_m$: molecular diffusivity of DO $[m^2/s]$

The phytoplanktonic process depends on phytoplankton respiration ($R_{O_2,pp}$) and photosynthesis ($P_{O_2,pp}$) as follows:

$$\frac{d[O_2]}{dt}_{phytoplanktonic} = P_{O_2,pp} - R_{O_2,pp} \tag{4}$$

And the bacterial process that is the main source of oxygen consumption depends on the heterotrophic bacterial kinetics and the availability of substrate matter (S, considered to be the rapidly biodegradable dissolved organic matter, $DOM_1$, in this model) as follows:

$$\frac{d[O_2]}{dt}_{bacterial} = -\tau_{HB}(1 - Y_{HB}) \; uptake \tag{5}$$

$$uptake = \frac{\mu_{max,HB} e^{-\frac{(T-T_{opt,HB})^2}{\sigma_{HB}^2}} \frac{[S]}{[S]+K_S}[HB]}{Y_{HB}} \tag{6}$$

where,

$[HB]$: the concentration of heterotrophic bacteria (hereafter, called bacteria) $[mgCL^{-1}]$

$\tau_{HB}$: 1.0 $[molO_2/molC]$ for full oxidation of OM in the respiration process

$Y_{HB}$: the growth yield of heterotrophic bacteria [-]

$uptake$: the uptake of substrate (here S = $DOM_1$) for bacteria growth $[mgCL^{-1}h^{-1}]$

$T_{opt,HB}$: optimum temperature for the growth of bacteria $[°C]$

$\mu_{max,HB}$: the maximum growth rate of bacteria at $T_{opt,HB}$ $[/h]$

$\sigma_{HB}$: standard deviation of bacteria temperature function $[°C]$

$K_s$: Monod half-saturation constant for bacterial growth (uptake constant) $[mgCL^{-1}]$

### 145 2.1.3 Organic matter degradation equations

The OM in the Seine originates from (i) point releases from WWTP and CSO, (ii) diffuse sources due to soil leaching and surface degradation through tributary rivers, and (iii) mortality of bacteria and phytoplankton (Billen et al., 2001). Figure 1 illustrates the OM related process of C-RIVE in the water column where the total organic matter (TOC) is initially divided into dissolved ($DOM$) and particulate ($POM$) forms. $DOM$ is composed of three different biodegradable fractions of (i) $DOM_1$





as the limiting substrate (rapidly biodegradable DOM in 5 days ), (ii) $DOM_2$ (slowly biodegradable DOM in 45 days), and (iii) $DOM_3$ (refractory DOM). Similarly, $POM$ is composed of (i) $POM_1$ (rapidly biodegradable POM), (ii) $POM_2$ (slowly biodegradable POM), and (iii) $POM_3$ (refractory POM). The benthic processes are not presented in Fig. 1.

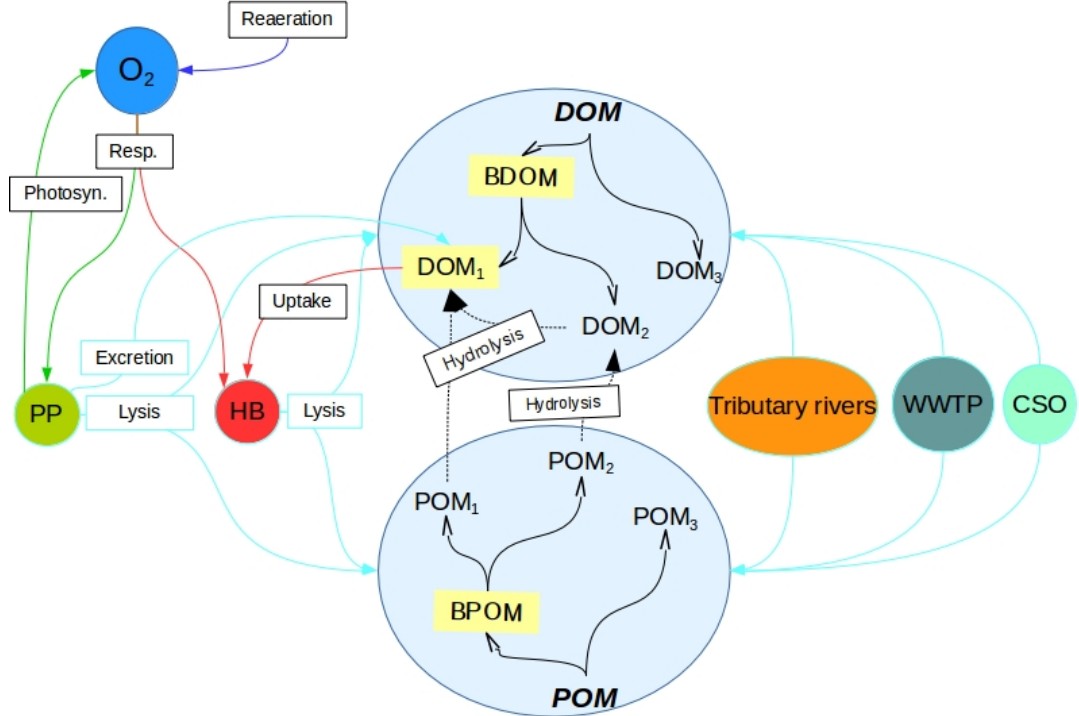

**Figure 1.** Schematic description of the OM-related process of C-RIVE in the water column. POM: particulate organic matter; DOM: dissolved organic matter; BDOM:biodegradable DOM, BPOM:biodegradable POM (subscripts 1, 2, and 3 refer to rapidly degradable, slowly degradable, and non-biodegradable fractions of OM, respectively); Cyan solid lines: OM input from sources and repartition between POM and DOM; Solid black lines: repartition of DOM and POM into biodegradability pools ; Dotted black lines: Hydrolysis; Remaining solid lines: Biogeochemical processes. Resp.:Respiration; Photo,:Photosynthesis; PP: primary producer; HB: heterotrophic bacteria; WWTP: wastewater treatment plant; CSO: combined sewage overflow

The degradation of OM happens through the uptake of small monomeric organic substrates ($S$, here $S = DOM_1$) by heterotrophic bacteria on the basis of the HSB model (Billen et al., 1988; Servais, 1989; Billen, 1991) and presented by Eq. (7) and

(9). These substrates are either the direct input ($P_S$) of $DOM_1$ from OM sources or produced from the exoenzymatic hydrolysis of the macromolecular fractions of both dissolved ($DOM_2$) and particulate ($POM_1$, $POM_2$) organic matter (Billen, 1991) or they originate from the phytoplankton excretion, which produces more easily utilizable OM ($DOM_1$) and microorganism lysis products that are macromolecular matter (Fig. 1) (Larsson and Hagstrom, 1979; Garnier and Benest, 1990; Billen, 1991).



$$\frac{d[S = DOM_1]}{dt} = hyd_{DOM_2} + hyd_{POM_{1,2}} - uptake_{HB} + P_S + P_E + P_L \tag{7}$$

where,

$P_S$, $P_E$, $P_L$: represent $DOM_1$ from the direct input of OM sources, phytoplankton excretion, and microorganism lysis, respectively $[mgCL^{-1}h^{-1}]$

$hyd_{DOM_2}$: hydrolysis of $DOM_2$ into $DOM_1$ based on the exoenzymatic hydrolysis equation of Michaelis-Menten $[mgCL^{-1}h^{-1}]$

$hyd_{POM_{1,2}}$: hydrolysis of $POM_1$ and $POM_2$ into $DOM_1$ and $DOM_2$, respectively, by first-order kinetics $[mgCL^{-1}h^{-1}]$


$$hyd_{DOM_2} = k_{hyd,max} \frac{[DOM_2]}{[DOM_2] + K_{DOM2}} [HB] \tag{8}$$

$$uptake_{HB} = \mu_{max,HB} \frac{[DOM_1]}{[DOM_1] + K_s} [HB] \tag{9}$$

where,

$uptake_{HB}$: uptake or consumption of $DOM_1$ by heterotrophic bacteria $[mgCL^{-1}h^{-1}]$

$k_{hyd,max}$: coefficient for hydrolysis of $DOM_2$ into $DOM_1$ $[/h]$

$K_{DOM2}$: constant of semi-saturation for the hydrolysis of $DOM_2$ $[mgCL^{-1}]$

### 2.2 Parameterization of organic matter share (repartition) and degradation

In order to account for the uncertainties related to the parameterization of OM degradation kinetics and its repartition into different constituent fractions, the following two sets of parameters are introduced:

#### 2.2.1 OM degradation parameters

C-RIVE parameters related to OM degradation are $K_s$ (represents uptake of $DOM_1$ by bacteria), $K_{DOM2}$ and $k_{hyd,max}$ (represent hydrolysis of $DOM_2$ to $DOM_1$), which have been defined in section 2.1.3. Hydrolysis parameters of POM are not considered in this study because the rate of hydrolysis of $POM_{1,2}$ is slower than that of $DOM_2$ by an order of magnitude of 100 to 1000(Billen et al., 1994).

#### 2.2.2 OM repartition or share parameters

C-RIVE does not include any parameter to define the repartition of OM into DOM and POM and then further into their corresponding fractions $DOM_{1,2,3}$ and $POM_{1,2,3}$; therefore, the following five parameters are introduced to represent the repartition of OM:

$$t = \frac{DOM}{TOC}$$





$$b_1 = \frac{BDOM}{DOM}$$

$$s_1 = \frac{DOM_1}{BDOM}$$

$\quad b_2 = \dfrac{BPOM}{POM}$

$$s_2 = \frac{POM_1}{BPOM}$$

where,

$TOC$: total organic matter or carbon (= $DOM + POM$) $[mgCL^{-1}]$

$\quad BDOM$: biodegradable DOM (= $DOM_1 + DOM_2$) $[mgCL^{-1}]$

$BPOM$: biodegradable POM (= $POM_1 + POM_2$) $[mgCL^{-1}]$

$t$: ratio between dissolved and total organic matter [-]

$b_1$: ratio between biodegradable DOM and DOM [-]

$s_1$: ratio between rapidly biodegradable DOM and biodegradable DOM [-]

$\quad b_2$: ratio between biodegradable POM and POM [-]

$s_2$: ratio between rapidly biodegradable POM and biodegradable POM [-]

However, these parameters are not inserted into C-RIVE; instead, the six OM fractions ($DOM_{1,2,3}$ and $POM_{1,2,3}$, which are C-RIVE inputs) are calculated in terms of the aforementioned five independent parameters based on the classification
presented in Fig. 2 and thereby they are reflected in the six model inputs.

### 2.2.3 Parameters for SA and bibliographical review of their variation ranges

Following parameterization, the 17 parameters as listed in Table 1 are evaluated in the SA. This includes two physical parameters that account for $O_2$ re-aeration; seven bacterial parameters that account for bacteria growth, mortality, and respiration; three OM degradation parameters that demonstrate OM kinetics; and five OM share parameters that represent the repartition
of TOC into smaller dissolved and particulate fractions.

Before proceeding to SA, it is necessary to specify the range of variation of these parameters according to the existing literature. As indicated in Table 1, the range of variation of the repartition and degradation parameters is selected based on a detailed bibliographical review, which is discussed and tabulated thoroughly in Hasanyar et al. (2020, 2021) , while the variation of physical and bacterial parameters is retrieved from Wang et al. (2018). Table 1 also includes the range of variation
of TOC, which represents the total organic matter input in the model due to the boundary conditions and varies from 1 to 10 mgCL[-1] under low flow (also retrieved from Hasanyar et al. (2020)). However, phytoplanktonic parameters are not included in this study because they exert an influence only during algal bloom periods (Wang et al., 2018), whereas this study is conducted under non-bloom situations where heterotrophic activity is dominant.




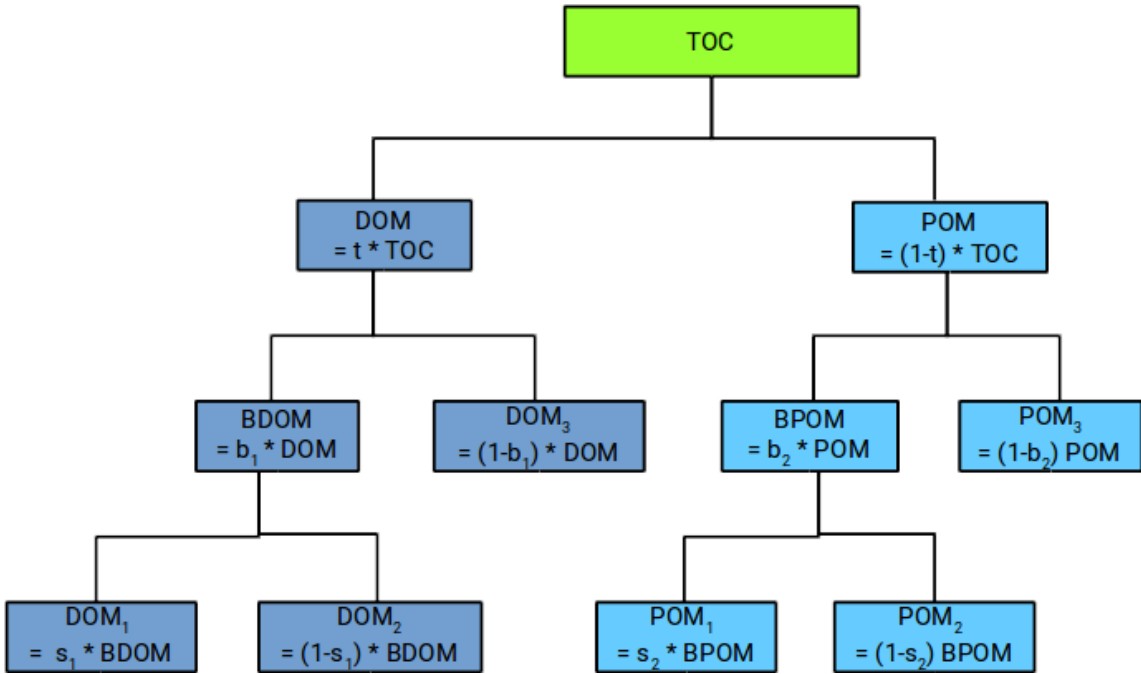

**Figure 2.** OM repartition into the six fractions of dissolved and particulate matter in terms of the five repartition parameters, namely, t, $b_1$, $s_1$, $b_2$, $s_2$

## 2.3 Case study

The synthetic case developed by Wang et al. (2018) was adapted for the application of SA methods on C-RIVE parameters during a low-flow period (Fig. 3). It is a stretch with a width of 100 m and a length of 1000 m representing the Seine. The low-flow period is identified with a discharge of 80 $m^3/s$ based on the data at Bougival station during the summer season. The simulation period is set at 45 days in order to be coherent with the experimental protocol of the BDOM measurement (Servais et al., 1995) where it is considered as the threshold between the biodegradable and refractory fractions of TOC in a

batch experiment. On the other hand, a 45-day simulation period is also necessary for studying the long-term effect of TOC degradation.

### 2.3.1 Initial conditions

A non-bloom low-flow situation (large heterotrophic bacteria biomass presence in the river) is considered to represent the low-flow period. Table 2 lists the initial concentrations for both water and sediment compartments that are set based on the mean

concentrations of the simulations at Bougival station during the 2007-2012 period (Vilmin et al., 2016) except for temperature (depending on summer season), DO (depending on oxygen solubility), POM and DOM fractions (depending on the TOC concentration and share parameters), and phytoplankton and bacterial biomass (depending on a post-bloom condition). Indeed,



**Table 1.** list of parameters and their corresponding ranges of variation

| Parameter | Description | Min. Val. | Max. Val. | Unit | References |
|---|---|---|---|---|---|
| **TOC** | Total organic carbon | 1 | 10 | [mgCL$^{-1}$] | |
| **OM share parameters** | | | | | |
| t | ratio between dissolved and total organic matter (DOM/TOC) | 0.4 | 0.9 | [-] | |
| $b_1$ | ratio between biodegradable DOM and DOM (BDOM/DOM) | 0.1 | 0.5 | [-] | |
| $b_2$ | ratio between biodegradable POM and POM (BPOM/POM) | 0.1 | 0.5 | [-] | |
| $s_1$ | ratio between rapidly biodegradable DOM and BDOM ($DOM_1$/BDOM) | 0.4 | 0.95 | [-] | Hasanyar et al. (2020, 2021) |
| $s_2$ | ratio between high biodegradable POM and BPOM ($POM_1$/BPOM) | 0.4 | 0.95 | [-] | |
| **OM degradation parameters** | | | | | |
| $K_s$ | constant of semi saturation for bacterial substrate uptake | 0.02 | 0.15 | [mgCL$^{-1}$] | |
| $K_{DOM2}$ | constant of semi saturation for the hydrolysis of $DOM_2$ | 0.2 | 1.5 | [mgCL$^{-1}$] | |
| $k_{hyd,max}$ | coefficient of the hydrolysis of $DOM_2$ to $DOM_1$ | 0.25 | 0.75 | [/h] | |
| **Bacterial parameters** | | | | | |
| $T_{opt,hb}$ | optimum temperature for bacterial growth | 15 | 30 | [°C] | |
| $\sigma_{hb}$ | standard deviation of temperature function for bacterial growth | 12.75 | 21.25 | [°C] | |
| $V_{sed,hb}$ | settling velocity of bacteria | 0 | 0.1 | [m/h] | |
| $K_{O_2,hb}$ | Half-saturation constant for dissolved oxygen | 0.375 | 0.625 | [mg$O_2$/L] | Wang et al. (2018) |
| $\mu_{max,hb}$ | maximum growth rate of bacteria | 0.01 | 0.07 [*] | [/h] | |
| $Y_{hb}$ | bacterial growth yield | 0.03 | 0.5 | [-] | |
| $mort_{hb}$ | bacterial mortality rate | 0.01 | 0.08 | [/h] | |
| **Physical parameters** | | | | | |
| $K_{navig}$ | re-aeration coefficient due to navigation | 0 | 0.05 | [m/h] | |
| $K_{wind}$ | re-aeration coefficient due to wind | 0.885 | 1.475 | [m/h] | |

[*] The upper limit identified by Wang et al. (2018) is decreased from 0.13/h to 0.07/h in order to avoid complete DO depletion in simulations longer than 5 days

the sum of $DOM_{1,2,3}$ and $POM_{1,2,3}$ is equal to the desired TOC, but they are distributed among the six fractions based on the five OM share parameters (t, $b_1$, $s_1$, $b_2$, $s_2$).





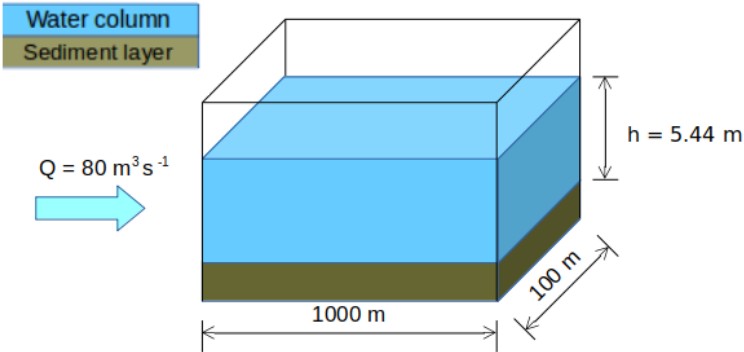

**Figure 3.** Synthetic scheme representing a reach of the Seine (modified from Wang et al. (2018))

**Table 2.** Initial concentrations of the simulations

| No | Species | $C_{ini,water}$ | $C_{ini,sediment}$ | Unit |
|---|---|---|---|---|
| 1 | $NH_4$ | 0.12 | 0.33 | [mgN/L] |
| 2 | $NO_2$ | 0.04 | 0.04 | [mgN/L] |
| 3 | $NO_3$ | 7 | 4.54 | [mgN/L] |
| 4 | TSS | 16.82 | 95010 | [mg/L] |
| 5 | $PO_4$ | 0.1 | 0.27 | [mgP/L] |
| 6 | $O_2$ | 8.62 | 6.65 | [mg$O_2$/L] |
| 7 | HB | 0.023 | 0.016 | [mgCL$^{-1}$] |
| 8 | PP | 0.010 | 0.003 | [mgCL$^{-1}$] |
| 9 | $DOM_1$ | f(TOC, share parameters) | 0.12 | [mgCL$^{-1}$] |
| 10 | $DOM_2$ | f(TOC, share parameters) | 1.28 | [mgCL$^{-1}$] |
| 11 | $DOM_3$ | f(TOC, share parameters) | 1.94 | [mgCL$^{-1}$] |
| 12 | $POM_1$ | f(TOC, share parameters) | 44 | [mgCL$^{-1}$] |
| 13 | $POM_2$ | f(TOC, share parameters) | 696 | [mgCL$^{-1}$] |
| 14 | $POM_3$ | f(TOC, share parameters) | 2555 | [mgCL$^{-1}$] |
| 15 | $T_{mean}$ | 22.4 ± 3.0 | | °C |

## 235  2.4  Sensitivity analysis strategy

The objective is to identify different scenarios under which the SA has to be conducted in order to detect the influential parameters under a summer low-flow condition. Therefore, a coarse pre-analysis consisting in forward simulations of the



C-RIVE model is first conduced with extreme values of a small number of representative parameters. Then, the necessary scenarios are developed to assess the assumptions and conclusions put in place in the pre-analysis.

### 2.4.1 Pre-analysis of the model under parameter extreme limits

First, we need to select certain parameters for the pre-analysis. We consider $\mu_{max,hb}$, $mort_{hb}$ and $Y_{hb}$ as they were found to be influential in the study of Wang et al. (2018) under non-bloom situations. However, to decrease the number of parameters, $mort_{hb}$ and $\mu_{max,hb}$ are represented together as a single parameter called "net growth (NG)."

$$Net\ Growth\ (NG) = \mu_{max,hb} - mort_{hb}$$

Fixing $mort_{hb}$ = 0.02/h at its reference value and $\mu_{max,hb}$ ranging between 0.022 and 0.07/h, net growth was found to range from 0.002 to 0.05/h while the range for $Y_{hb}$ is taken from Table 1. On the other hand, as the OM share parameters are not C-RIVE inputs, we consider BDOM to represent them in the model. Its range is calculated in Eq. (10)-(11) in order to be statistically independent based on the TOC repartition diagram (Fig. 2) as follows:

$$BDOM_{min} = TOC_{ref} * t_{ref} * b_{1,min} \tag{10}$$


$$BDOM_{max} = TOC_{ref} * t_{ref} * b_{1,max} \tag{11}$$

Here, $TOC_{ref}$ is a reference TOC value and fixed at 5 mgCL$^{-1}$ (considered as the baseline concentration of TOC in the Seine (Vilmin et al., 2016)), the reference t ($t_{ref}$) = 0.7 is the average value of $t$ variation range and $b_1$ is taken from Table 1.

**Table 3.** Combinations of the three parameter values for eight single simulations

| Sim. No. | BDOM | Net growth | $Y_{hb}$ |
|---|---|---|---|
| 1 | 0.35 | 0.05 | 0.03 |
| 2 | 0.35 | 0.002 | 0.03 |
| 3 | 1.75 | 0.05 | 0.03 |
| 4 | 1.75 | 0.002 | 0.03 |
| 5 | 1.75 | 0.05 | 0.5 |
| 6 | 1.75 | 0.002 | 0.5 |
| 7 | 0.35 | 0.05 | 0.5 |
| 8 | 0.35 | 0.002 | 0.5 |

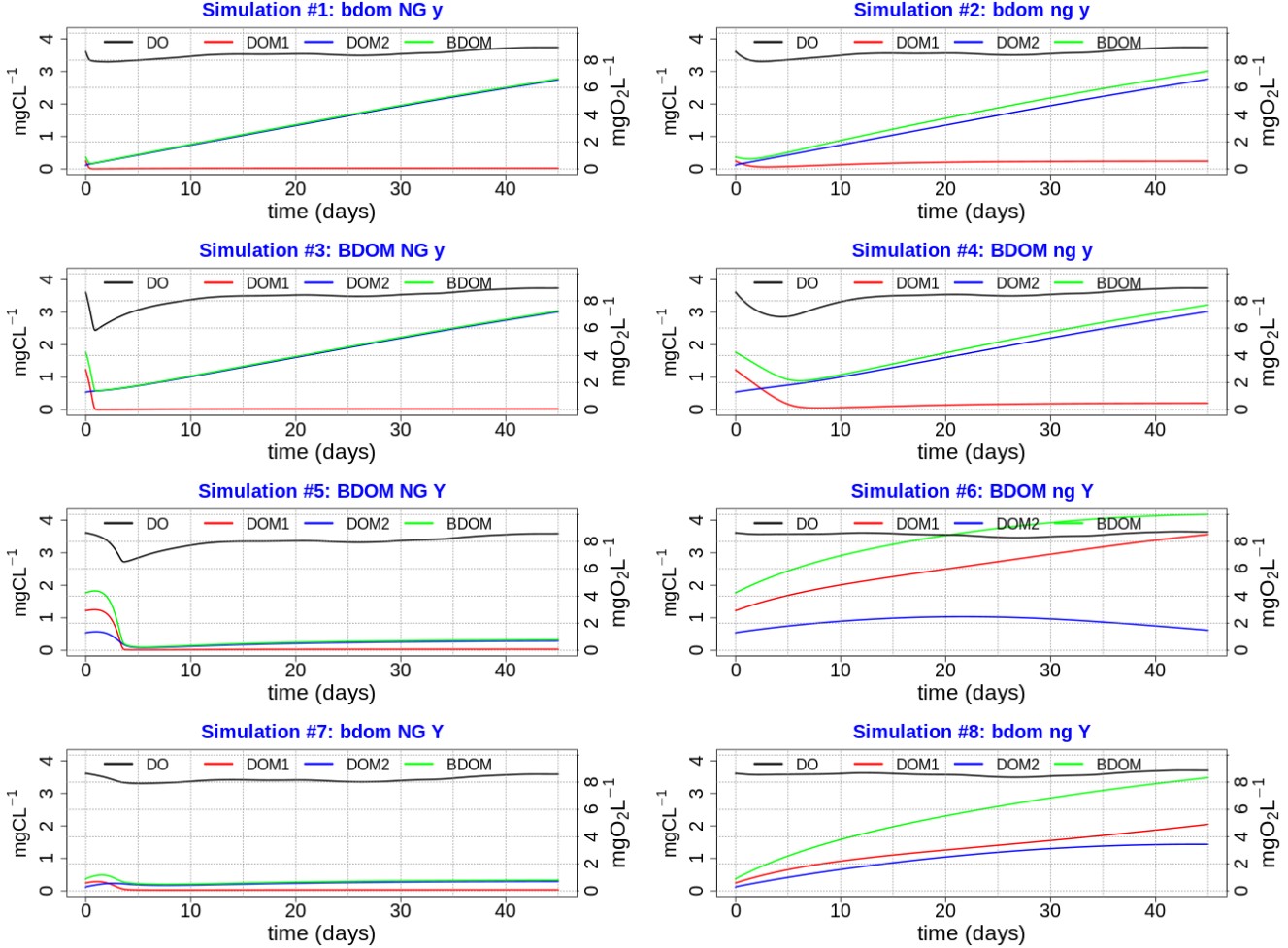

**Figure 4.** Eight plots of single simulations; (XX yy ZZ) Any parameter name written in capital or small letters means that its maximum or minimum value is used, respectively, in that specific single simulation. For example, plot #3 (BDOM NG y) that corresponds to simulation #3 in Table 3 is a simulation where the maximum values of BDOM and net growth and the minimum value of $Y_{hb}$ are used

Therefore, eight simulations pertaining to eight different combinations of the minimum and maximum values of these three
parameters are conducted (Table 3) and accordingly for each combination, the evolution of DO, $DOM_1$, $DOM_2$, and BDOM is plotted (Fig. 4). In order to discriminate easily among the eight plots of single simulations using their titles, any parameter name written in capital or small letters means that its maximum or minimum value is used, respectively in that particular simulation. For example, plot 3 (BDOM NG y), that corresponds to simulation 3 in Table 3, is a simulation where the maximum values of BDOM and net growth and the minimum value of $Y_{hb}$ are used.
As can be observed in Fig. 4, simulations 3 (BDOM NG y), 4 (BDOM ng y), 5 (BDOM NG Y) and 6 (BDOM ng Y) have high BDOM where all except simulation 6 demonstrate considerable DO depletion. This shows the importance of BDOM in





the depletion of DO. The reason for lack of depletion in 6 could be attributed to the combination of low net growth and a high yield due to which BDOM is not consumed. However, comparing simulations 6 (BDOM ng Y) and 7 (bdom NG Y), we observe that even a lower BDOM coupled with high net growth (simulation 7) has more effect on DO than a high BDOM coupled with low net growth (simulation 6). This shows the interaction effect of BDOM with net growth parameters and the fact that despite BDOM being the primary requirement for depletion of DO, the net growth needs to be high in order to demonstrate the influence of BDOM and provide the means for its consumption which would result in DO depletion. Moreover, the difference between two horizontally adjacent simulations is in the net growth, which is maximum for the left-hand simulations and minimum for those on the right and as a result, all simulations on the left side with a high net growth demonstrate more depletion or consumption of DO than those on the right side. This shows the influence of net growth parameters ($\mu_{max,hb}$, $mort_{hb}$) on the model at low flow.

### 2.4.2 Sensitivity analysis scenario development

Having performed the pre-analysis, we understood the importance of BDOM and that of the net growth parameters such that BDOM is needed primarily in order to be consumed so that DO could be depleted. And then we discern that BDOM consumption is high when net growth is at its highest value. Therefore, the following three different sensitivity analysis scenarios whose parameters are detailed in Table 4 need to be conducted.

The first SA is conducted by assuming the general influence of net growth parameters in the pre-analysis, and in order to have a broader view of the model sensitivity with respect to all the model parameters. Based on the pre-analysis, we observed that the main effect due to BDOM is linked to high net growth rates, therefore, we can assume that the effect of parameters other than net growth parameters is demonstrated when they are coupled with a high net growth condition. In addition, since a significant interaction (the difference between the first and total sensitivity indices) is observed between net growth parameters in Wang et al. (2018), they are assumed to be hiding the influence of other parameters. Therefore, in order to confirm these two assumptions and to observe the influence of other parameters, we implement a second SA where net growth parameters are fixed at its highest value. This SA removes the possibility of interaction among net growth and other model parameters. It results in a better evaluation of the model sensitivity with respect to the parameters whose influences might be hidden by the dominant and interacting parameters.

The third SA is performed to verify the second SA assumption that parameters other than net growth exert their influence only under a high net growth condition, and thus the same parameters could be deemed non-influential under a low net growth situation. therefore the net growth parameter is fixed at its lowest value. The settings for the three SAs are as follows:

1. **First SA (All parameters included)**: There are 17 defined parameters (Table 1 & Table 4) in the model, the simulation period is **45 days**. It is repeated 10 times for every TOC concentration of 1-10 mgCL$^{-1}$.

2. **Second SA (Fixed high net growth)**: The net growth parameters are fixed as follows:

$$Fixed\ high\ net\ growth = High\ bact.\ growth\ rate\ (\mu_{max,hb} = 0.07/h) - Bact.\ mortality\ rate\ (mort_{hb} = 0.02/h)$$





Furthermore, to decrease the computational cost of the model, the three OM share parameters (t, $b_1$ and $b_2$) from the first
SA are narrowed to BDOM and BPOM whose variation ranges are calculated based on the following Eq. (12)-(15) as
follows:

$$BDOM_{min} = TOC * t_{ref} * b_{1,min} \tag{12}$$

$$BDOM_{max} = TOC * t_{ref} * b_{1,max} \tag{13}$$

$$BPOM_{min} = TOC * (1 - t_{ref}) * b_{2,min} \tag{14}$$

$$BPOM_{max} = TOC * (1 - t_{ref}) * b_{2,max} \tag{15}$$

Here TOC varies between 1-10, therefore, similar to the first SA, this SA is repeated 10 times corresponding to each
TOC case. The pre-analysis also demonstrated that BDOM or precisely the substrate ($DOM_1$) is consumed in less than
5 days under the high net growth condition (simulations 1, 3, 5 & 7) , therefore imposing a **5 days** simulation period.
Twelve parameters are evaluated under this scenario (Table 4).

3. **Third SA (Fixed low net growth)**: This SA is conducted in a similar way to the second SA except that this time $\mu_{max,hb}$
   is fixed at a lower value of 0.022/h in order to simulate a very low net growth rate condition as follows:

   $$Fixed\ low\ net\ growth = Low\ bact.\ growth\ rate\ (\mu_{max,hb} = 0.022/h) - Bact.\ mortality\ rate\ (mort_{hb} = 0.02/h)$$

   Similar to the second SA, this SA is also repeated 10 times and implemented under a **5-day** simulation period. Twelve
   parameters are also evaluated under this scenario (Table 4).

## 2.5 Sensitivity analysis methodology

Each of the three aforementioned SAs is implemented based on an innovative SA methodology initially proposed in Wang et al.
(2018) and adopted in this study, where the influence of input parameters (X) is evaluated on the C-RIVE model according to
the variations of a large set of DO simulations (model output, Y). The followings steps are pursued in this approach:

1. **Input parameter identification**: Initially, a set of D input parameters (Table 4) are identified with their corresponding
   ranges of variation (Table 1).

2. **Parameter sampling and model input creation**: Saltelli's extension of the Sobol sequence (Saltelli, 2002) implemented
   in PYTHON SALIB package (Herman and Usher, 2017) is employed to create different combinations of the input





**Table 4.** The parameters considered in each of the four sensitivity analyses

|  | $1^{st}$ SA | $2^{nd}$ SA | $3^{rd}$ SA |
| --- | --- | --- | --- |
| OM share parameters | t | BDOM | BDOM |
|  | $b_1$ | BPOM | BPOM |
|  | $s_1$ |  |  |
|  | $b_2$ |  |  |
|  | $s_2$ |  |  |
| OM degradation parameters | $K_s$ | $K_s$ | $K_s$ |
|  | $K_{DOM2}$ | $K_{DOM2}$ | $K_{DOM2}$ |
|  | $k_{hyd,max}$ | $k_{hyd,max}$ | $k_{hyd,max}$ |
| Bacterial parameters | $T_{opt,hb}$ | $T_{opt,hb}$ | $T_{opt,hb}$ |
|  | $\sigma_{hb}$ | $\sigma_{hb}$ | $\sigma_{hb}$ |
|  | $V_{sed,hb}$ | $V_{sed,hb}$ | $V_{sed,hb}$ |
|  | $K_{O_2,hb}$ | $K_{O_2,hb}$ | $K_{O_2,hb}$ |
|  | $Y_{hb}$ | $Y_{hb}$ | $Y_{hb}$ |
|  | $\mu_{max,hb}$ |  |  |
|  | $mort_{hb}$ |  |  |
| Physical parameters | $K_{navig}$ | $K_{navig}$ | $K_{navig}$ |
|  | $K_{wind}$ | $K_{wind}$ | $K_{wind}$ |
| total number of parameters | 17 | 12 | 12 |

parameters, which are designed to produce optimized simulations and efficient analysis results. Considering a sample size of 10,000 (N) (needed for stable results based on Nossent et al. (2011)), a matrix with a size of N(2D+2) × D is created for each SA scenario where every row represents one set of input parameters for the model.

3. **Model simulation**: In this step, the model inputs are launched into C-RIVE for the simulation period considered with a 1-min time step. As an output, a DO time series matrix with a size of $N(2D+2) \times M$, where M is the number of output time steps based on a 15 min output time step [1], is created corresponding to each input matrix created in the previous step. Figure 5 demonstrates the ensemble of 260,000 [=N(2D+2)] DO simulations of TOC = 5 mgCL$^{-1}$ in the second SA scenario (TOC = 5 mgCL$^{-1}$ is used in this study to represent the TOC range of 1-10 mgCL$^{-1}$ and to show the results in case they are similar across all TOC concentrations).

---

[1] M = 45- or 5-day simulation period × 24 hrs × 3600 min / (1-min simulation time step × 15-min output time step) = 4230 or 480 depending on the simulation period, respectively





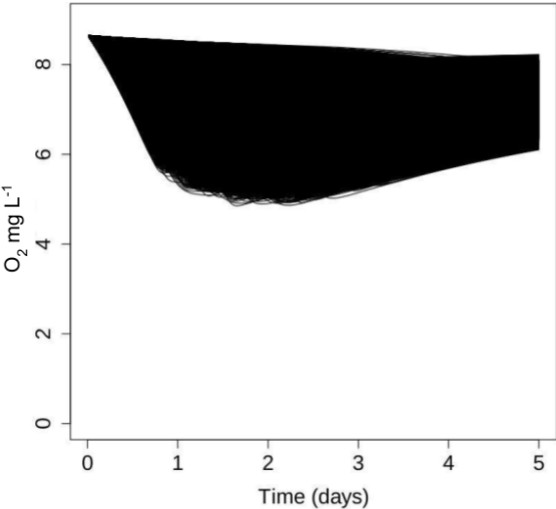

**Figure 5.** Ensemble of the 260,000 DO simulations for TOC = 5 mgCL$^{-1}$ in the second SA scenario

4. **Dimensionality reduction**: The empirical orthogonal function (EOF) method is an adaptation of principal component analysis (PCA) to study a phenomenon that changes with a continuous variable, such as time, and is applied to transform the output data from one coordinate system into another by introducing new uncorrelated (orthogonal) variables (principal components) (Jolliffe and Cadima, 2016). EOF is adopted to transform the model output, which is a DO times series matrix composed of M columns into a smaller matrix where each simulation can be represented by a linear combination

of EOFs. The coefficients of this linear combination are indeed orthogonal projections that maximize the variance while transforming the data from a higher-dimensional space into a lower one. The way EOF decreases dimensionality is such that it ranks the components based on the maximized variance. In other words, most of the information is kept in the first few components, thereby making it possible to reduce the number of dimensions without losing a considerable amount of information (Wold et al., 1987). In this study, the first $k$ EOF elements that constitute at least 99% of the total model

variance are considered to represent each single simulation of the DO time series as shown for TOC = 5 mgCL$^{-1}$ in the second SA (Fig. 5a), where four ($k$) significant EOFs are found such that the first EOF ($EOF_1$) represents almost 55% of the total variance. Figure 5b illustrates the evolution of the eigenvalues of the four (k) EOFs with time, which are consequently used to represent each simulation in terms of the $k$ new coordinates . Thereby, an $N(2D+2) \times M$ matrix is converted into a new matrix of $N(2D+2) \times k$, which will be subjected to the Sobol SA method. The R $prcomp$

function is used to conduct the EOF analysis.

    5. **Sobol sensitivity analysis**: The Sobol SA method (Sobol, 1993; Saltelli et al., 2010) is applied in this study to evaluate the sensitivity of the model output against the input parameters. It is a variance-based method that classifies the parameters based on their contribution to and/or influence on the total variance of the model output (Brookes et al., 2015). It is a convenient method to be used for SA of complex models that involve interactions between parameters. In this method,





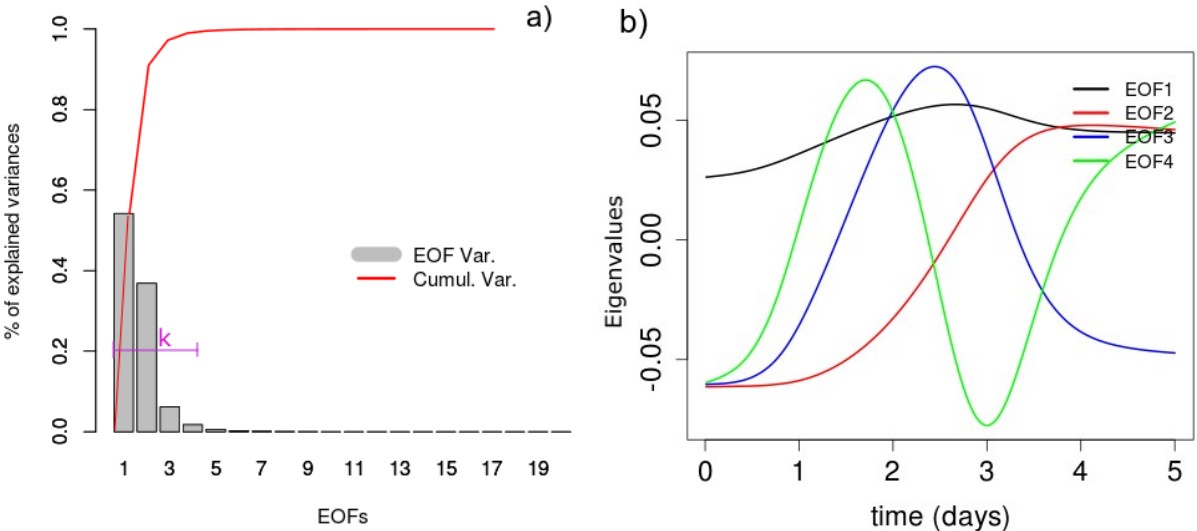

**Figure 6.** a) Cumulative sum of EOF variances and b) time evolution of four ($k$) significant EOFs for TOC = 5 mgCL$^{-1}$ in second SA

the model output (Y) is expressed as a function of D parameters:

$$Y = f(X) = f(X_1, ..., X_D),$$  (16)

such that the model output could be decomposed by elementary functions:

$$f(X) = f_0 + \sum_{i=1}^{D} f_i(X_i) + \sum_{i=1}^{D-1} \sum_{j=i+1}^{D} f_{ij}(X_i, X_j) + .... + f_{1,...,D}(X_1, ..., X_D)$$  (17)

Here $f_0$ is the expectation of the model output and each one of the elementary functions have a zero mean and can be
computed by integration:

$$f_0 = \int_{[0,1]^D} f(X)d_X$$  (18)

$$f_i(X_i) = -f_0 + \int_{[0,1]^{D-1}} f(X)d_{X\sim i}$$  (19)

$$f_{ij}(X_i, X_j) = -f_0 - f_i(X_i) - f_j(X_j) + \int_{[0,1]^{D-2}} f(X)d_{X\sim(ij)}$$  (20)

On the other hand, the total unconditional model variance could be defined as:

$$V(Y) = \int_{[0,1]^D} f^2(X)d_X - f_0^2$$  (21)





Thereby, the total unconditional variance of the model can be expressed as:

$$V(Y) = \sum_{i=1}^{D} V_i(X_i) + \sum_{i=1}^{D-1}\sum_{j=i+1}^{D} V_{ij}(X_i, X_j) + .... + V_{1,...,D}(X_1, ..., X_D) \tag{22}$$

where, $V_i$ is the partial variance of the $i_{th}$ parameter and $V_{ij}$ is the interaction effect of the $i_{th}$ and $j_{th}$ parameters. The partial variance is calculated as:

$$V_{i_1,...,i_s} = \int_0^1 ... \int_0^1 f^2_{i_1,...,i_s}(X_{i_1}, ..., X_{i_s})dX_{i_1}, ....dX_{i_s} \tag{23}$$

where $s = 1, ..., D$ and $f_i$ is an elementary function. Therefore, the first-order Sobol SA indices can be computed as follows:

$$S_i = \frac{V_i}{V} \tag{24}$$

$S_i$ is also called as the "main effect" because it represents the contribution of a single input parameter $i$ on the total variance. The total sensitivity index ($S_{Ti}$), also called "global effect," is another index that represents the sum of the first-order index ($S_i$) and the effect of the interaction between the parameters and is calculated as follows:

$$S_{Ti} = S_i + \sum_{j \neq i} S_{ij} + ... \tag{25}$$

here, $S_{ij} = \frac{V_{ij}}{V}$ is called the "second-order index" and measures the interaction between a pair of parameters $X_i$ and $X_j$. Therefore, the sum of second-order interactions of any parameter $X_A$ with other parameters ($X_B, ..., X_D$) is considered to represent the second-order index of each parameter ($S_2$) as follows:

$$S_{2,A} = \sum_j S_{Aj} \tag{26}$$

Since the output of previous step is a matrix of *k* vectors corresponding to the *k* EOFs, the Sobol indices of parameters are initially calculated *k* times for each EOF and then added while being weighted by the variance of the corresponding EOF.

The total computation time from step 1 to 5 is 12 h for each TOC case of the first SA, after using a parallel code in PYTHON and dividing the simulations in several groups to decrease the computational cost using 20 processors (Intel(R) Xeon(R) E5-2640 and frequency of 2.40GHz). The computation time is 3 h for each TOC case of the second and third SA, which clearly demonstrates the gain in time compared with the first SA.

## 3 Results

This section presents the results of the three Sobol SAs during a summer low-flow period. The influential parameters of each analysis are discussed in the following paragraphs.




## 3.1 First SA: All parameters

Figure 7a presents the results of the Sobol SA method on TOC = 5 mgCL$^{-1}$. It is expressed by a bar plot of the total sensitivity ($S_T$), first-order sensitivity ($S_1$), and second-order sensitivity ($S_2$) indices of the parameters. The higher-order sensitivity indices are also calculated in terms of the difference between the total and the first- and second-order indices ($S_T$-$S_1$-$S_2$). The parameters are ranked based on their $S_T$ and the most influential parameters are shown by the shaded area, which includes parameters constituting 95% of the total variance of the model output. However, this SA is conducted on different TOC values

of 1-10 mgCL$^{-1}$, and therefore in order to summarize all of the results together, the evolution of the normalized total sensitivity indices ($S_T^*$) of the six most influential parameters with TOC is presented in Fig. 7b.

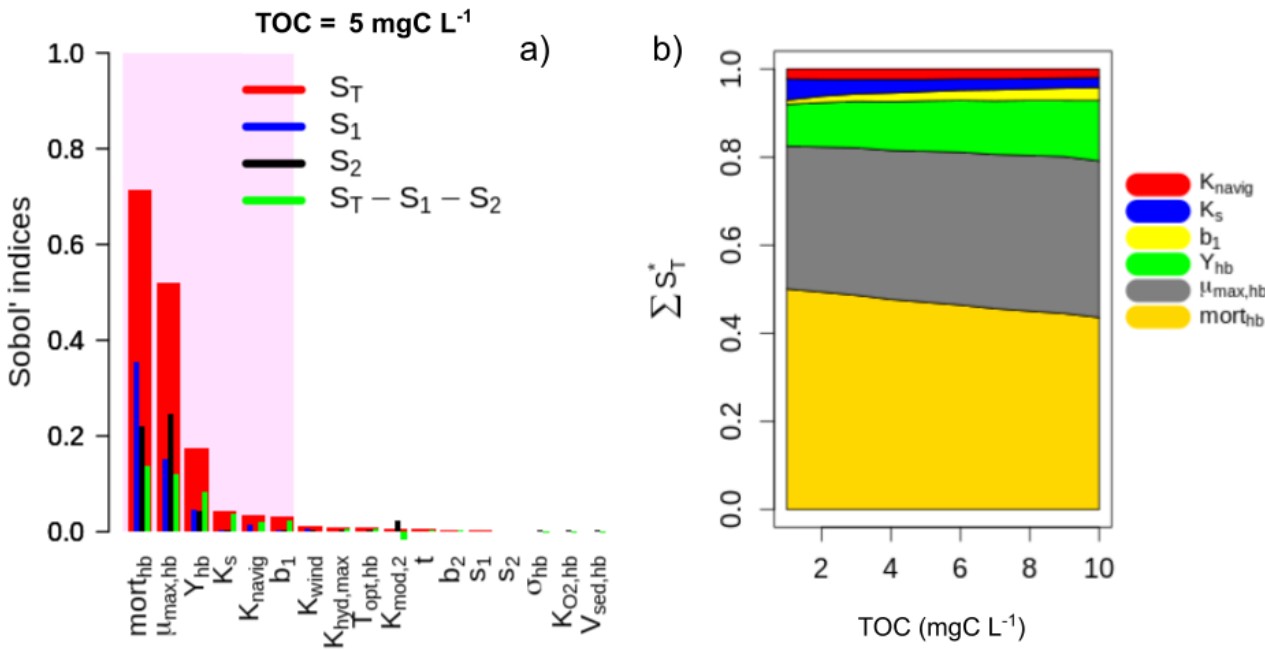

**Figure 7.** Sobol SA results of **first SA: All parameters** (a) Sobol SA results for TOC = 5 mgCL$^{-1}$; (b) Evolution of the normalized total sensitivity indices of the influential parameters with TOC

According to Fig. 7b, the three bacterial parameters of bacterial mortality rate ($mort_{hb}$), maximum bacterial growth rate ($\mu_{max,hb}$), and bacterial yield ($Y_{hb}$) exert the most influence on DO evolution, and whatever the TOC concentration, they represent the majority of the model sensitivity. By increasing TOC, we observe a gradual decrease in the influence of $mort_{hb}$,

but an increase in the influence of $Y_{hb}$. This result obtained over the 45-day simulation period (Fig. 7) confirms the assumption made in the pre-analysis step (sec 2.4.1) regarding the overall dominance of bacterial parameters in long-term low-flow periods and is in accordance with the findings of the study by Wang et al. (2018), which was conducted over a 4-day simulation period.





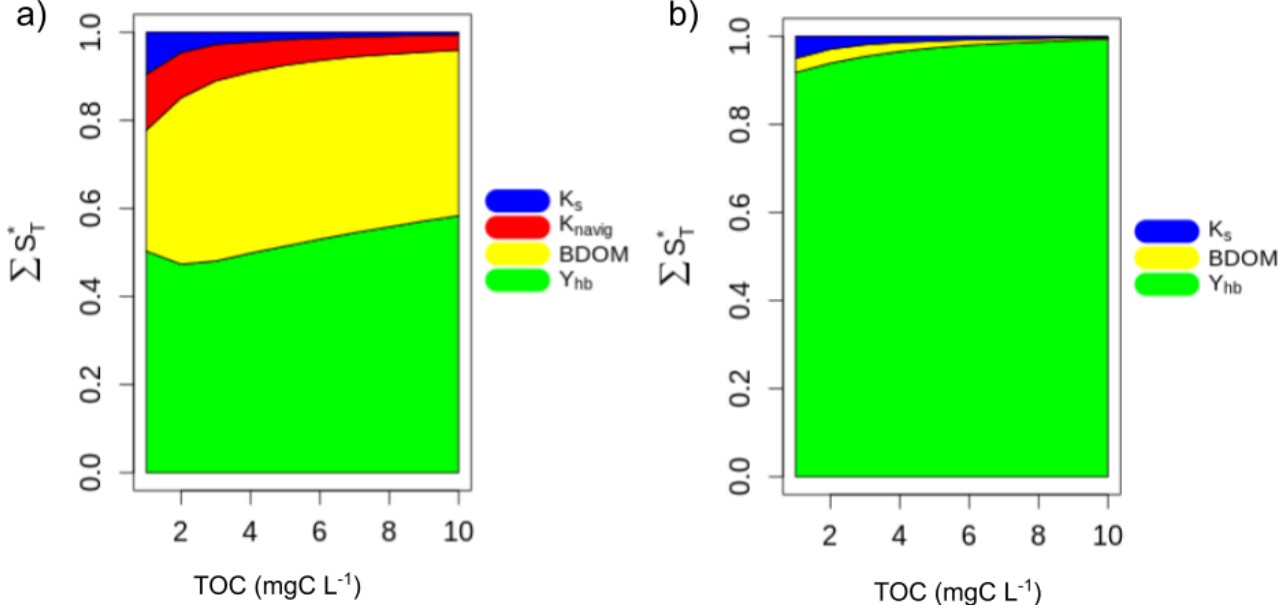

**Figure 8.** Results of (a) Second SA: Fixed high net growth; (b) Third SA: Fixed low net growth

Apart from the constant of navigation ($K_{navig}$), which is a physical parameter, the other two influential parameters ($K_s$, $b_1$) are OM-related parameters that were introduced in this study. $K_s$ seems to be more important in lower TOC concentrations compared to $b_1$ whose influence increases in higher TOC concentrations.

On the other hand, these results confirm the assumptions made in the pre-analysis step that the dominant parameters tend to hide the influence of other parameters. Observing the second-order ($S_2$) and higher-order sensitivity ($S_T$-$S_1$-$S_2$) indices of $mort_{hb}$, $\mu_{max,hb}$ and $Y_{hb}$ in Fig. 7a, very strong interactions can be highlighted between these parameters, i.e., a significant portion of their total sensitivity indices is due to their internal interactions. On the other hand, $b_1$ and $K_s$ also exert an influence due to their higher-order interaction with these three parameters. Thus, in order to see what happens behind net growth parameters, and in order to observe the dominant parameters under two extreme conditions of high and low net growth rates, the second and third SAs are conducted by fixing the $mort_{hb}$, $\mu_{max,hb}$ parameters under the notion of net growth rate.

Moreover, the three OM parameters (t, $s_1$ and $s_2$) are found to be non-influential, which means they can be excluded from SA by fixation in the second and third SA while calculating the variation ranges of BDOM and BPOM (Eq. (10)-(13)). Finally, the inclusion of $b_1$ among influential parameters out of the five OM parameters confirms the selection of the BDOM concentration instead of other OM components in the pre-analysis step.





### 3.2 Second SA: Fixed high net growth

Based on Fig. 8a which depicts the results of the second SA, it is shown that $Y_{hb}$ and BDOM are the most influential parameters under high net growth rate conditions. This is due to the fact that the bacterial community manages to consume most of BDOM
under a high net growth condition and then at some point, BDOM becomes a limiting factor for their growth. This result confirms the assumption made in the pre-analysis that the influence of parameters other than net growth parameters will be displayed if they are studied under a high net growth condition. The other important parameters are $K_{navig}$ and $K_s$ whose influence is reduced by increased TOC. Moreover, very small interactions are observed between the parameters because almost all of their global influence stem from their main effects ($S_T \approx S_1$ for each parameter), which once again confirms the previous
consideration that interactions are related to the effect of a varying net growth rate.

### 3.3 Third SA: Fixed low net growth

The results of the third SA (Fig. 8b) reveal that $Y_{hb}$ is a predominantly influential parameter under a low net growth rate condition across all TOC concentrations. This is due to the fact that the bacterial community hardly grows at all and therefore BDOM is not a limiting factor for bacterial growth as there is not enough bacterial activity under a low net growth rate
condition. This result verifies the assumption made in the second SA by displaying all previous influential parameters except $Y_{hb}$ as non-influential.

In observing the role of BDOM, we conclude that by increasing the net growth rate, the influence of BDOM increases. Here, we need to highlight that the actual upper limit of net growth is 0.11/h (Wang et al., 2018) compared to the current value of 0.05/h because the upper limit of $\mu_{max,hb}$ is reduced in Table 1 in order to prevent complete DO depletion of simulations that
occur due to the combination of extreme values in Sobol sampling. Therefore, it can be envisaged that BDOM may have a greater influence under the actual range of net growth rate.

### 3.4 The most influential parameters during a summer low-flow period

This study confirms that over a 45-day summer low-flow period and whatever the TOC concentration (first SA), the bacterial net growth rates represented by $mort_{hb}$, $\mu_{max,hb}$ and $Y_{hb}$ are predominant and have the most influence on the DO evolution.
On the other hand, since a significant portion of the total sensitivities of the bacterial parameters arise from inter-parameter interactions, by fixing both the $mort_{hb}$ and $\mu_{max,hb}$ under the concept of a fixed net growth rate once at a maximum value to prohibit their interactions with other parameters (second SA) and by studying the river over a shorter period of 5 days, BDOM and $Y_{hb}$ appear to be influential parameters as well. It highlights the importance of a proper characterisation of BDOM during summer low flow periods, when heterotrophic respiration is at its highest level due to a high water temperature. This illustrates
that BDOM is the most influential fraction of the total organic matter entering a river system from its boundary conditions.





## 4    Discussion

The results of this study highlight the importance of having a better knowledge of i) organic matter outflow from boundary conditions during low flow, and ii) in particular of its biodegradable portion (BDOM), iii) in addition to bacterial physiology for water quality modeling at low flow. Therefore, these results can be expected to have the following consequences on data
assimilation and water quality monitoring in urban areas:

### 4.1    Consequences of the results on data assimilation (DA) strategy

Inclusion of BDOM and other influential parameters from this study in data assimilation tools is crucial for improving their performance at low flow. Despite its early development in freshwater sciences, the usage of assimilation techniques remain limited among the scientific community (Cho et al., 2020; Wang et al., 2021). Since the 1970's, implementations of assimilation
techniques switched from extended Kalman filter (EKF) (Beck and Young, 1976; Beck, 1978; Bowles and Grenney, 1978; Cosby and Hornberger, 1984; Whitehead and Hornberger, 1984; Pastres et al., 2003; Mao et al., 2009) to ensemble Kalman filter (EnKF) (Beck and Halfon, 1991; Huang et al., 2013; Kim et al., 2014; Page et al., 2018; Zhang et al., 2020; Loos et al., 2020), which remains today the most popular assimilation method (Carrassi et al., 2018; Cho et al., 2020) that mostly focuses on phytoplankton dynamics in rivers and lakes or situations other than a low flow where DO is the state-variable.

On the other hand, in the first application of a particle filter for data assimilation in limnology using ProSe-PA software to study a dry year (Wang et al., 2019; Wang, 2019; Wang et al., 2021), mismatches were found between simulations and observation data in low-flow periods. This is due to the degeneracy effect of the low flow data assimilation tool which cannot reach oxygen concentrations as low as those observed in the natural environment and it is considered to be due to insufficient biodegradable organic matter loading in the model caused by underestimated BDOM inputs to the Seine river (Wang et al.,
2021). Therefore, we propose the incorporation of BDOM (the most influential OM-related parameter through the $b_1$ parameter) as a new component of the particle filter implemented in ProSe-PA to facilitate its estimation and consequently improve the simulation results. Consistent with the results of the second and third SA, it would be appropriate to fix one or both of the $mort_{hb}$ and $\mu_{max,hb}$ parameters in order to decrease the parameter interactions and thereby better characterize the distribution of BDOM and other model parameters under low flow. In addition, to facilitate the inclusion of BDOM in any data assimilation
tool, it is necessary to explicitly include the model of OM sharing (Fig.2) as part of the software, which will read TOC as an input data from now on.

      Moreover, BDOM from each organic matter source (tributary river, WWTP, and CSO) should be independently represented in the DA scheme due to its distinct role and contribution of organic matter in urban rivers (Servais and Garnier, 1993; Seidl et al., 1998b). The fact that the effluents from WWTP and CSO contain large bacteria which have significantly different
physiological properties than the small autochtonous ones (Garnier et al., 1992; Servais and Garnier, 1993) also advocate for the identification of input data following the three aforementioned pools. To represent the contrasted concentrations of bacteria upstream and downstream metropolis such as Paris urban area, two bacterial communities should be taken into account in the forward model. A strategy consisting of reducing the degree of freedom of the mathematical problem should also be





implemented in the DA framework such as fixing a few parameters values of each community, for instance, their growth and
mortality rates assuming that the net growth of small bacteria is slow while that of large bacteria is fast.

## 4.2 Consequences of the results on water quality monitoring in urban areas

The results of this study suggest the implementation of a densified (optimally designed) water quality monitoring system using
high-frequency sensors to measure the pertinent information necessary for improving the model uncertainties and monitoring
of the river metabolism at low flow. In addition, these results propose regular monitoring of bacterial populations to detect
any potential bacterial community shift following infrastructure developments in WWTP and transient events such as CSO.
Therefore, continuous monitoring is essential for overall ecosystem management and for studies of human impacts (Beaulieu
et al., 2013).

### 4.2.1 How to densify the urban monitoring networks?

Considering the importance of the contribution of organic matter from boundary conditions during low-flow periods, we pro-
pose the establishment of monitoring stations within a denser monitoring network capable of characterizing the upstream
tributary rivers and the outflow of anthropogenic sources such as WWTP and CSO because an appropriate spatial resolution
(Polus et al., 2010) and consideration of point and non-point pollution sources (Dixon et al., 1999; Do et al., 2012) is necessary
in the design of monitoring networks. Ouyang (2005) used principal component analysis (PCA) to identify the necessary loca-
tions for establishing a new station or removal of previous stations in addition to specifying DOM and TOC among the main
important variables that should be monitored.

This new constraint on the development and densification of monitoring networks is in addition to the classically proposed
pragmatic criteria such as the (i) existence of pollution sources and variability in water quality between the existing monitoring
stations, (ii) infrastructure for ease of accessibility to the station, and (iii) the location of tributaries to the system (Chilundo
et al., 2008; Anvari et al., 2009) and less frequently the spatial variability of processes (Polus et al., 2010). Meanwhile, Strobl
et al. (2006) have considered the information on land use such as point and non-point outflow sources and the watershed soil
characteristics, hydrology, and topography in order to select the monitoring points; generally, there is no correlation between
the size and the number of monitoring stations (Nguyen et al., 2019). Another practical way is to observe the discontinuities or
jumps in DO profiles obtained from modeling software, which may help in determining the location of monitoring stations.

### 4.2.2 How important are high-frequency data in water quality modeling?

We propose the collection of high-frequency BDOM data from boundary conditions as being indispensable in improving the
model uncertainties and detecting the concentration peaks. Indeed, the hydro-biogeochemical processes take place on different
time scales ranging from minutes and hours to weeks and months (Tomlinson and Carlo, 2003; Vilmin et al., 2018). In order
to be able to study both the trajectories of the systems (long term) (Flipo et al., 2021) and the transient events such as storm
overflows, it is necessary to maintain high frequency monitoring networks over time. Such networks allow us to follow the





influence of WWTP discharges and also the influence of storm overflows which provoke longitudinal decreases of downstream $O_2$ due to the direct relation between bacterial respiration and the increase of the BDOM (Stanley et al., 2012). Therefore, the conventional sampling campaigns followed by incubation experiments (Servais et al., 1995) for determining BDOM cannot meet the need for models and water managers. Thus, the sampling frequency should be set according to the variability of the desired variable.

However, the operational and economic aspects of any monitoring system should be optimized so as to keep the costs at a minimum while meeting the monitoring requirements (Dixon and Chiswell, 1996; Camara et al., 2020). Therefore, optimal or minimum sampling frequencies have been suggested for different water quality variables depending on the presence of outflows and their corresponding contribution to the river, the degree of acceptable uncertainty in the model, the monitoring goals, and the variability of the target variable itself (Chappell et al., 2017; Vilmin et al., 2018).

### 4.2.3 Monitoring of bacterial communities is necessary for detection of community shift

Considering the results of this study, which introduce several bacterial parameters (mortality and growth rates, yield) as having the most influence on DO evolution, we propose the implementation of regular bacterial community sampling campaigns in order to provide updated information on the physiological properties of bacterial communities in the case of an autochtonous community shift and modifications of waste water treatment processes. Samples should thus be taken in river waters upstream

from the major urban areas, and in the effluents of major WWTPs and CSOs. This issue arises because, on the one hand, rivers are highly sensitive to urban outflows at low flow (Seidl et al., 1998a; Huang et al., 2017) due to their low dilution capacity and, on the other hand, because with the construction of a new WWTP or any infrastructure development inside a WWTP as well as following transient events such as overflow from a CSO, the outflow quality changes in terms of bacteria, DOM, and BDOM concentrations (Servais and Garnier, 1993; Seidl et al., 1998b). As a result, this induces potential shifts in the bacterial

community, which have been found to be related to DOC source and its biodegradability (Hullar et al., 2006; Crump et al., 2003) such that an increase in BDOM is considered to increase the diversity of bacterial populations (Landa et al., 2013); therefore, one could recommend the need for regular reassessment of the influential bacterial parameters (Even et al., 2004). Wang et al. (2019); Wang (2019); Wang et al. (2021) have also considered the lack of information on heterotrophic bacterial communities as another source of mismatch in modeling at low flow. Indeed, here we can propose that the sampling frequency

in the monitoring stations must now be considered not only according to the temporal variability of the variable of interest, but also according to the possible successions of species. Moreover, such updated data on bacterial communities would also help in narrowing down the variation ranges of the parameters, which would improve water quality model performances, especially of data assimilation softwares.

Finally, we believe that once a water quality model capable of data assimilation is validated and its uncertainties are suitably

reduced, it can provide acceptable estimates of water quality variables in periods when monitoring is not possible or at locations where accessibility is an issue (Reis et al., 2015; Jiang et al., 2020). The coupling between water quality monitoring networks and particle filter modeling softwares will lead to a very powerful generation of water quality spatio-temporal interpolators that will be able to estimate water quality state indicators with a unpreceeded resolution and accuracy. In other words, if





the approach followed by PIREN-Seine (https://www.piren-seine.fr/en) research program since its beginnings has shown all

its power (Carré et al., 2021) via the determination of the properties of the species communities by laboratory incubation experiments (Servais et al., 1995), it appears today that the coupling between measurement networks and modeling makes it possible to approach the functioning of the systems in a more dynamic way by the identification of the communities parameters that vary with time (Wang et al., 2021). This would lay the foundation for using the model in forecast studies (Park et al., 2020) and for assessing different river management strategies (Reis et al., 2015).

**5   Conclusions**

The objective of this work was to investigate the role of organic matter loadings to river systems and the physiology of bacteria in river metabolism during a summer non-bloom low-flow period. New parameters were introduced to account for repartition and degradation of OM. Then, the sensitivity of the C-RIVE model was analyzed against the newly introduced and the already existing model parameters. The following conclusions can be drawn from this study:

– First, the Sobol sensitivity analysis method proved very efficient in the identification of influential parameters on DO evolution in the C-RIVE model. Then, by fixing the interaction-inducing parameters, the influence of other parameters was assessed;

– The net growth rate of bacteria composed of their maximum growth rate ($\mu_{max,hb}$) and mortality rate ($mort_{hb}$) is the most important parameter under different TOC concentrations; therefore, it is essential to have a better estimation of the

variation ranges of the growth and mortality rates of bacterial communities;

– Model response is very sensitive to the biodegradable share of DOM (BDOM) contributed by the boundary conditions. The effect of this parameter prevails at higher net growth rates occurring during summer low-flow periods when the organic matter attributable to human pressure is abundant in the river;

– The river metabolism is dominated by bacterial activity at low flow during summer non-bloom periods;

– Water quality monitoring networks should be densified to encompass all important BDOM contributing boundary conditions. BDOM data should be collected at high frequency;

– More frequent sampling of autochtonous bacteria communities upstream and downstream of major urban areas and in majors WWTP and CSO effluents will be of considerable interest to validate time varying parameter values estimated by data assimilation frameworks;

– The results of this study provide us with a list of influential and non-influential parameters. The latter can be fixed at their average or preferred value as per the literature, and the former can be introduced to the data assimilation tools in order to estimate their temporal evolution thanks to the high-frequency data observed, such as DO.





*Code availability.* Name of software : C-RIVE

Contact address: nicolas.flipo@mines-paristech.fr

Year first available: 2019

Program language: ANSI C

Operating system: Linux

Software access: Deposit in progress

Availability and cost: Open source

Licence: Eclipse public licence under discussion

*Author contributions.* The author contributions are as follows:

The calculations were designed by Masihullah Hasanyar, Thomas Romary and Shuaitao Wang;

The calculations were executed by Masihullah Hasanyar;

The results were analyzed by all authors;

The first draft of the article was drafted by Masihullah Hasanyar, and then it was revised and corrected by all authors;

And finally Nicolas Flipo did the search for funding;

*Competing interests.* This work is a contribution to the PIREN-SEINE research program, part of the french Long-Term Socio-Ecological Research (LTSER) site "Zones Ateliers Seine". The authors would like to thank SIAAP and SEDIF for their collaboration and we declare that there are no known conflicts of interest associated with this work and there has been no significant financial support for this study that
could have influenced its outcome. The authors alone are responsible for the content and writing of this article.

*Acknowledgements.* This work is a contribution to the PIREN-SEINE research program, part of the french Long-Term Socio-Ecological Research (LTSER) site "Zones Ateliers Seine."





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
