# Peer review of "How much do bacterial growth properties and biodegradable dissolved organic matter control water quality at low flow?"

_Biogeosciences, 2021_

## Author Response (AR1)

**Author's response**

Dear editor,

Please find hereafter our point by point reply to the reviewers' comments, including the changes brought to the manuscript mentioned by their line numbers in the manuscript pdf file.

**Response to reviewer #1**
First of all, we would like to thank the anonymous reviewer for their kind attention to our article and the time that they allocated to read it  and provided us with comments and suggestions for enhancing our paper. Hereafter are our answers to the 4 comments raised by the reviewer.

Comment #1:
It is not clear whether the sensitivity results will hold if there will be source-sink terms for organic matter.
Response #1:
Each of the three sensitivity analyses are conducted with varying total organic matter (TOC) sources between 1 to 10 mgC/L (Fig. 7-8). Therefore, the source term is indeed considered and adding a new source will not change the results.

As for organic matter sink, since we had designed the study under a situation where the system shall not be depleted of organic matter and dissolved oxygen at any moment, as it is our goal to see how the parameters behave in their presence, this study cannot stand under a sink term that would lead to depletion, especially of organic matter. Indeed, in a depleted environment, the influence of the organic parameters cannot be studied.

To summarize, the fact that we conducted many numerical experiments under various organic matter conditions already answers the reviewer's concern. The answer to the effect of a poor or rich environment in organic matter is already largely discussed throughout the paper. This limitation and other upcoming restrictions of this study are addressed in a separate subsection in the revised manuscript under Discussion part, section 4.2, lines 470-474.

Comment #2:
No attention is paid to the radiation effects of the bacteria population that is most pronounced at low flows
Response #2:
We would like to thank you for bringing up this point. Indeed, our model currently lacks this process and it is something which needs to be incorporated in the model source code. However, its implementation and reanalysis of the work will take far more time than the review period of this article. Moreover experimental data on the subject needs to be found in order to conduct a sensitivity analysis. This specific question may be the subject of further research. We discussed this limitation of our approach in lines 475-480 of the manuscript.

Comment #3:

The role of the hyporheic exchange in bacteria population dynamics at low flows is ignored and can be substantial.

Response #3:
The contribution of groundwater to downstream rivers is well known to be negligible with respect to the discharge of those rivers (Strahler order > 6). Our case study mimics such rivers, and especially the Seine river crossing the Paris urban area. For such a system, Pryet et al. (2015) provide estimates of aquifer contribution to the Seine River. More specifically, the hyporheic exchange rate is very limited in this area, with a maximum value of 0.005 m3.s-1.km-1, which corresponds to a maximum of 1 m3.s-1 over a 200 km stretch of downstream river. With respect to the actual discharge of the Seine river of 80 m3.s-1, this process is not relevant to account for in our study. However the reviewer is correct that if studying headwater streams functioning, the hyporheic exchanges would be of primary importance. The negligence of hyporheic exchanges is mentioned in the initial conditions section lines 250-252.

Comment #4:
The introduction of constant parameters to simulate the repartitioning is a gross simplification. Monitoring data show that the ratios vary.

Response #4:
Without any doubt, the organic matter partitioning parameters vary with time. However, from a sensitivity analysis point of view, we need to keep them constant during the simulation period so that the influence of their increment could be studied on the model total variance and then ranked using the Sobol criteria. That is why we have 360,000 (N(2D+2)) parameter combinations where each parameter value is changed within its variation range given in Table 1 but unchanged during the simulation period to study their impact on model output. Fortunately, the concept of time varying parameters has been addressed in our data assimilation study (https://doi.org/10.1016/j.envsoft.2022.105382) which is the second step after the sensitivity analysis as it has been explained in discussion subsection "Consequences of the results on data assimilation (DA) strategy". In that work, we had estimated the time evolution of parameters thanks to observation data and a new work is under way to test an automatic detection of boundary conditions organic matter content using data assimilation. To our knowledge this would be a major step forward for the modeling of the water quality of downstream river systems.

**Response to reviewer #2**

First of all, I would like to thank the reviewer for their positive comments on our paper and valuable propositions for improvement. Here are our subsequent responses, clarifications and the modifications that we added in the manuscript:

*In response to their comment:*
*"What is the novelty this manuscript brings to the field?"*
**[R1]** Our article provides the following novelties compared to the previous works and are summarized in the introduction part of the manuscript in lines 65-69.:

1. In the previous works, Wang et al., (2018) have focused on both high & low flow under different conditions of bloom, non-bloom and post-bloom, whereas here we have explored further the summer post-bloom low flow condition where significant discrepancies are observed between water quality model results and dissolved oxygen (DO) observations (Wang et al., 2022). We already knew that bacterial physiological parameters control DO evolution during low flow and that was already quantified in Wang et al., (2018). However, what we didn't know is the extent to which the characteristics of the organic matter (OM), particularly its dissolved biodegradable fraction, control the oxygen dynamics at low water levels, and whether these characteristics are important with respect to the physiological properties of heterotrophic bacteria;

2. To explore this question through a sensitivity analysis, we had to explicitly add the parameters of OM model inside the software CRIVE itself, and especially in the boundary conditions of the model, where CRIVE used to read the six pools of OM (DOM1, DOM2, DOM3, POM1, POM2, POM3) as time varying state variables defined by the user. We therefore added an OM repartition model inside CRIVE so that the DOMi and POMi state variables are now calculated by the model based on only one value provided by the user, i.e. the total organic carbon (TOC), and 5 new model parameters (t, b1, b2, s1, s2). This repartition model not only distributes TOC among the six CRIVE pools using the 5 parameters whose variation ranges were found using a bibliography review (Table 1) , but it also gives us the possibility to do a sensitivity analysis for evaluating their role and influence on DO variation. The main difference is that instead of reading the 6 pools (not varying) directly, now it reads TOC and uses the 5 parameters to convert it into the 6 pools (now varying because the 5 parameters have a range).

3. Compared to the previous work (Wang et al., 2018) that has studied only the direct impact of each parameter, we went further and looked into intra-parameter interactions (higher order Sobol indices) and we found that certain parameters hide the influence of other parameters due to their interactions. Thanks to that, we designed the 2nd and 3rd Sobol' sensitivity analysis which allowed us to better quantify how the share of OM influences DO in river systems with regards to the physiological parameters of heterotrophic bacteria. We believe that this methodology may also be of interest for future sensitivity analysis where parameter interactions may hide the effect of other parameters.

4. In the previous study, the sensitivity analysis was conducted under a constant OM load of TOC =3.2 mgC/L whereas in this work, we evaluated the evolution of sensitivity indices for various TOC loadings ranging from 1 up to 10 mgC/L which represents the OM load from river, treatment plants and combined sewage overflows (Fig. 7b and Fig 8).

5. Conducting long-term sensitivity analysis: In the previous work, the influence of model parameters are usually studied over a short period of time, for instance a 4 day period in Wang et al. (2018). Here we looked deeper inside the system dynamics by extending the period up to 45 days. Such strategy led to a better understanding of the mid-term effect of slowly biodegradable OM. Even though those effects appear rather negligible, this result is important for improving and simplifying water quality models.

The reviewer advises to make the findings clearer in the paper. We agree with this comment and propose to add a first section to the discussion to wrap up those findings as stated here after.

*In response to their comment:*
*"The importance of heterotrophic bacteria activity and properties of the dissolved organic matter pool are pinpointed as important parameters to explain uncertainties of water quality models in the introduction (lines 49 to 60). Then, what is this manuscript offering new (or different) from previous studies? "*

**[R2]** The reviewer was misled by a flawn formulation of this section of the introduction. We rephrased it following those clarifications in lines 50-54. Formerly, Wang et al., (2022) simply assumed from their study that the OM degradation and OM repartition are playing a role in the model discrepancies during low flow, without explicitly quantifying their relative influences. In our paper we tested those hypotheses and therefore extended the parameters of interest to include 3 parameters representing OM kinetics and 5 representing OM repartition to quantify the sensitivity of DO variation with respect to those with a Sobol sensitivity analysis. We found that b1, the share of BDOM, has a significant influence on DO variations in certain circumstances, such as the presence of fast growing heterotrophic bacteria. In that case, a low b1 value may lead to a depletion of BDOM by heterotrophic bacteria, while high b1 allows the micro-organism to grow without limit, leading to significant oxygen depletions.

*In response to their comment:*
*"Please, specify better what are the research questions or objectives of this work?"*
**[R3]** We added the following main research questions in the introduction of the article in lines 70-76:
- What are the influential parameters controlling DO during a post-bloom summer low flow period where discrepancies are observed in different water quality models? Is a model that includes bacteria physiological parameters only sufficient to describe DO variation ?
- To what extent is the knowledge of the quantity of OM share, especially that of BDOM influential for water quality modeling?
- What is the hierarchy among the influential parameters ?

*In response to their proposal on the discussion part:*
**[R4]** We totally overhauled and restructured the discussion section by first answering the research questions with the current section 3.4. This new section is called "Hierarchy of the most influential parameters during low flow period".
Then, we discussed the limitations and assumptions of this study as the second subsection as indicated in the response to the comments of the first reviewer. We also provided recommendations for future studies in order to incorporate these limitations lines.

Then, we restructured the sub section "Consequences of the results on water quality monitoring in urban areas" by reformulating how important are our results in the context of water quality

monitoring and what information or experimental data is required to be supplied to the water quality models in order to provide better estimates of the river water quality. Indeed, we will first show how important it is to have better identification of bacterial parameters in any water quality monitoring network and second what we can do to get more information on b1 or BDOM. As recommended by the reviewer, we removed the parts related to the design of monitoring stations.

The last subsection of the discussion is dedicated to data assimilation where more clarifications are made for technical terms such as data assimilation and particle filter and also suggestions on how to conduct the data assimilation based on our results

*In response to their second question regarding the incorporation of new parameters:*
**[R5]** We restructured the material and method section 2.2.2 lines 195-220 to display how we have incorporated the organic matter repartition model consisting of 5 new parameters inside CRIVE instead of using the 6 forced user inputs that are not model parameters.

*In response to their specific questions on the addition of new parameter like:*
*What do authors mean with new parameters? New regarding what exactly? C-RIVE?*

**[R6]** Here, we have two types of new parameters. First, OM degradation kinetic parameters that already exist in CRIVE but whose influence was not studied in any other research. Secondly, OM repartitioning parameters (section 2.2.2) that as I have explained in the point #2 of R1**.** This is a novelty that did not exist in CRIVE before. Indeed, CRIVE used to read the share of each one of the 6 OM pools directly as an input (that was not variable, they were created in a form of database by multiplying TOC with certain assumed values), however, what we did as a novelty was that we gave CRIVE the possibility to read directly TOC (which comes from experimental data) and convert it into the above 6 OM pools using 5 parameters for which we did an extensive bibliography to find their variation range (t,b1,b2,s1,s2). Thereby, we created these 5 new parameters whose influence on DO could be studied and thanks to which we can now have varying 6 OM pools. This is something which was not possible before. This is clarified in lines 216-222.

*In response to the question: Another change is that authors pooled DOM1 and DOM2 fractions to create a new fraction called BDOM. Am I missing something? What is really new/different in this approach regarding to previous work in C-RIVE?*
**[R7]** If we look at equations 12 & 13, the variation range of BDOM is found using the variation range of b1, therefore it is not pooled by addition of DOM1 and DOM2. On the contrary, DOM1 and DOM2 are now derived from BDOM using the parameter s1. But why did we do this? In the first sensitivity analysis, we found b1 as an influential parameter, however for the second and third Sobol and in order to decrease the computation cost, we used BDOM as a parameter to get rid of the 5 initial parameters (as shown in Table 4, we went from 17 parameters to 12 parameters). BDOM is the equivalent of b1 (b1 = BDOM/DOM). DOM is constant (DOM = t x TOC) because t was found to be non-influential in the first experiment and fixed here, therefore, having b1 or BDOM does not make any difference technically.

**[R8]** *Finally, regarding the use of repetitive and introductory paragraphs*, we think that it is a good habit to brief the reader regarding what they are going to expect in different sections of an article. This will give them the chance to fast access to their intended sections or subsections. Therefore, we believe that the short briefing of the article at the end of introduction or a fast description of the method section in its first paragraph is a good habit in modeling articles which helps the reader better understand how the different elements of the method section are interlinked. However, we will find the annoying sentences and will remove or rephrase them.

As required by the editor, clear definitions and clarifications were added for technical terms such as data assimilation and particle filter in order to facilitate a wider audience in lines 506-509 and 510-511, respectively. Finally, necessary modifications were made to the figures in order to make them compatible with the color blindness requirements.

References:

Pryet, A., Labarthe, B., Saleh, F., Akopian, M., and Flipo, N. (2015) Reporting of stream-aquifer flow distribution at the regional scale with a distributed process-based model, Water Resour. Manage., 29, 139-159. doi: 10.1007/s11269-014-0832-7

Wang, S., Flipo, N., and Romary, T.: Time-Dependent Global Sensitivity Analysis of the C-RIVE Biogeochemical Model in Contrasted Hydrological and Trophic Contexts, Water Research, 144, 341–355, https://doi.org/10.1016/j.watres.2018.07.033, 2018.

Wang, S., Flipo, N., and Romary, T.: Oxygen Data Assimilation for Estimating Micro-Organism Communities' Parameters in River Systems, Water Research, 165, 115 021, https://doi.org/10.1016/j.watres.2019.115021, 2019.

Wang, S., Flipo, N., Romary, T., and Hasanyar, M.: Particle Filter for High Frequency Oxygen Data Assimilation in River Systems, Environmental Modelling & Software, 151, 105382, 2022 https://doi.org/10.1016/j.envsoft.2022.105382

---

## Author Response (AR2)

**Author's response**

Dear editor,

First of all, we would like to thank the editor for their kind attention to our article and the time that they allocated to read it and provided us with comments and suggestions for enhancing our paper.
Please find hereafter our point by point reply to comments, including the changes brought to the revised manuscript mentioned.

1) Please include some lines of reasoning why and how a purely model-based approach is actually capable of addressing the research questions. The paper does not include comparison of whatever model output is created with even a single observed data point. This is a severe limitation that should be acknowledged right from the beginning and then again in the discussion.

Response: C-RIVE is a C implementation of the RIVE model. RIVE model was developed in the 90's (Billen et al. 1994, Garnier et al., 1995). The model is community centered and explicitly describes micro-organisms such as phytoplankton and heterotrophic bacteria. The physiological parameters of those communities were determined through multiple lab experiments. Both the model and its parameterisations were coupled in two river water quality models : RIVERStrahler and ProSe, which were both validated on real case applications in multiple river basins over the world such as in the Mosel river (Germany) (Garnier et al., 1999), in the Scheldt river (Belgium and Netherlands) (Billen et al., 2005, Thieu et al., 2009), in the Day-Nhue river (Vietnam) (Luu et al., 2021), in  the Seine river (France) (Raimonet et al., 2015, Vilmin et al., 2015, Vilmin et al., 2016 ,Garnier et al., 2020), in the Somme river (France) (Thieu et al., 2009, Thieu et al., 2010), in the red river system (China and Vietnam) (Quynh et al., 2014), in the Danube river (Romania and Bulgaria) (Garnier et al., 2002), in the Zenne river (Belgium) (Garnier et al., 2013), and in the Lule and Kalix rivers (Sweden) (Sferratore et al., 2008).

This is added in the CRIVE subsection 2.1 of Methods.

 2) The paper is very long and not very accessible. This has two reasons: First, some text can simply be shortened without much loss of information. I make a few suggestions below (there are several more places) where I urge you to search for briefer expressions. Second, the technical sections are very long. I have already suggested to move parts of the methods into a supplementary document. You may deem this inappropriate. If so, please provide a justification.

Thanks for the two proposals on shortening the paper. First we rephrased and shortened the paragraphs based on your suggestions below and at other parts as well. Then, we shortened the methods section by moving some of its parts to the supplementary material and rephrased its paragraphs. This included equations already published in other articles and those equations proposed in this study but were somehow redundant in the article. We also combined Fig. 1 and 2 into one figure demonstrating all related processes. The results and discussion parts were

similarly reworded. As a result, the number of pages (abstract to conclusion) reduced from 26 to 18 pages.

In response to your specific comments. Lines numbers correspond to that of the track change manuscript:

33: too long sentence that should be broken apart: (34: done)

38. Sentences that can clearly be shortened due to redundant content. Whole paragraph is overly long. (41: whole paragraph is shortened and reworded)

50: Reword, double negation in "lack" and "inability"? Sentence unclear. The whole paragraph needs to be shortened. (55: whole paragraph is shortened and reworded, the message is made clear)

58: Please shorten paragraph, partly redundant. Last sentence is too long. (63: shortened and reworded)

60: Please reconsider the word "repartition". This feels very French to me, maybe better "partitioning"? Applies throughout the manuscript. You later also use the word "share". (repartition replaced with partitioning in all over the manuscript)

69: Whose functioning? (81: corrected to "the functioning of the Sobol sensitivity analysis")

70: Rather "inter-parameter"? Shorten sentence. Be more specific about "hiding effect". (81: shortened and reworded accordingly)

71: Rather "simulation period" (73: reworded the whole paragraph)

73: Please express briefer. Here "We address three research questions" would also do it without any information loss. (84:edited accordingly)

75: Suggest "parameters for bacterial physiology" (87: growth and yield rates) suggested

78: "," missing after "(BDOM)" (90, edited accordingly)

79: meaning of "hierarchy" is unclear (91: importance ranking) added

81: meaning of "against the background of.." is unclear. (removed)

83: Why "Finally,"? (the whole paragraph is merged with the paragraph before the research questions)

89: First sentence superfluous. Shorten whole paragraph, please. (101: whole paragraph shortened)

90: I think the "goal" of a study cannot be to "use a method". Please reword. (103: whole paragraphed shortened and reworded)

136/137: Please provide units for K_rea. I think the proper expression for both D_s and K_rea would be as a "(vertical) velocity" and not a (dimensionally unclear) "coefficient". (edited accordingly and moved to the supplementary material line 12/13)

163: what is "CSO"? (It is the abbreviation of combined sewer overflow as mentioned in line 34, in the caption of figure 1 and in )

102: what is a particle filter? Maybe better to first introduce ProSe, then ProSe-PA. There are some lines of explanations in the discussion (around line 599) that could be moved here. (The ProSe-PA subsection is removed during the overhauling of the methods section because we are modeling using C-RIVE which is one of the libraries of ProSe-PA. So the particle filter remains defined at the previous location)

280: Move information to figure legend. (It was already mentioned in the figure legend as well, so to avoid repetition, removed from the text)
622: This really does not seem to be a sound and safe conclusion to draw from this study! There is absolutely zero comparison to observation data. (We agree, this paragraph was an opinion about future implications of data assimilation after adding BDOM among assimilable parameters, therefore, we removed it from the text as it doesn't add value to the text)

Figure 1: the "cyan" coloring will hardly be printable and remain visible. (A new figure combining figures 1 and 2 is created where the cyan coloring changed to blue)
Figure 3: Maybe not really needed? Or consider as a sub-panel in Figure 1? (Actually, we need to keep it because it helps some readers who are interested in figures to better understand our case study).

References

Billen, G., Garnier, J., and Hanset', P.: Modelling Phytoplankton Development in Whole Drainage Networks: The RIVERSTRAHLER Model Applied to the Seine River System, Hydrobiologia, 289, 119–137, https://doi.org/10.1007/BF00007414, 1994

Billen, Gilles, Josette Garnier, and Véronique Rousseau. "Nutrient fluxes and water quality in the drainage network of the Scheldt basin over the last 50 years." Hydrobiologia 540.1 (2005): 47-67.

Garnier, J., Billen, G., and Coste, M.: Seasonal Succession of Diatoms and Chlorophyceae in the Drainage Network of the Seine River: Observation and Modeling, Limnology and Oceanography, 40, 750–765, https://doi.org/10.4319/lo.1995.40.4.0750, 1995

Garnier, J., Billen, G., and Palfner, L. "Understanding the oxygen budget and related ecological processes in the river Mosel: the RIVERSTRAHLER approach." Man and river systems. Springer, Dordrecht, 1999. 151-166.

Garnier J., G. Billen, E. Hannon, S. Fonbonne, Y. Videnina, M. Soulie, "Modelling the Transfer and Retention of Nutrients in the Drainage Network of the Danube River.", Estuarine, Coastal and Shelf Science, 54 (3), (2002), 285-308,

Garnier, Josette, et al. "Modeling historical changes in nutrient delivery and water quality of the Zenne River (1790s–2010): The role of land use, waterscape and urban wastewater management." Journal of Marine Systems 128 (2013): 62-76.

Garnier, Josette, et al. "Ecological functioning of the Seine River: From Long-term modelling approaches to high-frequency data analysis." The Seine River Basin. Springer, Cham, 2020. 189-216.

Le Thi Phuong Quynh, Garnier Josette, et al. "Preminary results of riverstrahler model application to the red river system (Vietnam)." Journal of Chemistry 47.1 (2014): 110-115.

Luua, Minh TN, et al. "Water quality in an urbanized river basin impacted by multi-pollution sources: From comprehensive surveys to modelling." SCIENCEASIA 47.1 (2021): 86-+.

Raimonet, Mélanie, et al. "Modelling the fate of nitrite in an urbanized river using experimentally obtained nitrifier growth parameters." Water research 73 (2015): 373-387.

Thieu, Vincent, Gilles Billen, and Josette Garnier. "Nutrient transfer in three contrasting NW European watersheds: the Seine, Somme, and Scheldt Rivers. A comparative application of the Seneque/Riverstrahler model." Water research 43.6 (2009): 1740-1754.

Thieu, Vincent, Josette Garnier, and Gilles Billen. "Assessing the effect of nutrient mitigation measures in the watersheds of the Southern Bight of the North Sea." Science of the total environment 408.6 (2010): 1245-1255.

Vilmin, Lauriane, et al. "Pluri-annual sediment budget in a navigated river system: the Seine River (France)." Science of the Total Environment 502 (2015): 48-59.

Vilmin, Lauriane, et al. "Carbon fate in a large temperate human‑impacted river system: Focus on benthic dynamics." Global Biogeochemical Cycles 30.7 (2016): 1086-1104.

---

## Author Response (AR3)

**Author's response — bg-2021-333**

Dear editor,

First of all, we would like to thank the editor for their kind attention to our article and the time that they allocated to read it and provided us with technical corrections for enhancing our paper. Please find hereafter our point by point reply to comments, including the changes brought to the manuscript.

In response to your minor technical corrections, lines numbers correspond to that of the production manuscript:

line 14: unclear meaning of phrase "and post-infrastructure improvement in treatment plants". We reworded it to "modifications in internal processes of treatment plants", line 14. Indeed a change in the efficiency or the internal processes of a treatment plant results in a modification of the outflows (bacteria and organic matter). This condition may lead to dominance of new bacteria communities in a river, that is why we had mentioned that a new bacterial community monitoring is necessary in such conditions.

15: Last sentence of abstract has unclear meaning. (It was reworded to "Furthermore, we discuss the inclusion of BDOM in statistical water quality modeling software for improvement in the estimation of organic matter inflow from boundary conditions", line 15.)

58-62: Q1 and Q2 seem to be not cleanly separated, consider rephrasing. Also in Q1 it may better be "alone" instead of "only" before "sufficient to describe DO variation". "Alone" is corrected, however, we believe the research questions are well separated because Q1 is answered in the results of first SA (section 3.1) where we discuss the general behavior of parameters and especially that of bacterial parameters. In Q2, which is answered in section 3.2 & 3.3, we focus on the role of BDOM under high and low net growth conditions.

99: unclear sentence, consider rephrasing. Reworded to "Using this OM partitioning model and depending on these five parameters, we are able to convert time varying TOC of boundary conditions, per say river inflows, into time varying $DOM_{1,2,3}$ and $POM_{1,2,3}$ fractions.". Line 99.

114-116: I do not understand why this would be relevant. The water does not need 45 days to flow through the modelled reach. As explained in line 118, it is a lagrangian approach, where we follow a river body along a river network of the above mentioned dimension with a speed of 0.14 m/s. Thus, there is no outflow in the system. We just move together with a 1000 meter long reach of the river. However, regarding the 45 days, we need to wait so long for the organic matter in the system to be consumed by the bacteria depending on the amount of organic matter and bacteria kinetics, thereby, resulting in a decrease in the dissolved oxygen levels as shown by figure 3. This is needed in order to evaluate the influence of parameters on the variation of oxygen. In other words, it takes around 5 days for the most labile part ($DOM_1$) to

be uptaken. Then we need to wait until day 45 for the hydrolysis of less labile (DOM_2) and particulate fractions into DOM_1.

169: correct to "their highest value".( Done, line 169)

212-213: not a sentence, maybe just delete "As" at start of sentence? (Done, line 212)

249: correct to "to synthesize our results". (Done, line 249)

267: Meaning of carbon sink unclear to me. The model has sinks. Or do you mean complete disappearance of carbon? By carbon sink we mean complete depletion or disappearance of carbon as explained in line 267. "Indeed, carbon depletion …. " . The carbon sink term was brought up by the first reviewer and therefore we included it as a limitation of the study.

270: What are "radiation effects"? Are you referring to sunlight? Yes. This was raised by the first reviewer as well. To make it clear for readers, we replaced it with "impact of solar radiation on" in line 270, and added the word "solar" before the word "radiation" in lines 271 and 274.

277: Typo in "diluation". (corrected, line 277)

334: Typo in "majorS... effluents" (corrected, line 334)